# Revealing the Driving Factors of Water Balance in Lake Balkhash Through Integrated Attribution Modeling

Ruibiao Yang<sup>1,2,3</sup>, Jinglu Wu<sup>1,2</sup>, Guojing Gan<sup>1,2</sup>, Ru Guo<sup>1,2,3</sup>

<sup>1</sup>State Key Laboratory of Lake Science and Environment, Nanjing Institute of Geography and Limnology, Chinese Academy of Science, Nanjing 210008, China

<sup>2</sup>University of Chinese Academy of Sciences, Beijing 100049, China

<sup>3</sup>College of Nanjing, University of Chinese Academy of Sciences, Nanjing 211135, China

Correspondence to: Jinglu Wu (w.jinglu@niglas.ac.cn)

Abstract. Understanding the impacts of climate change and human activities on large endorheic lakes is crucial for sustainable water management, yet quantitative attribution remains a significant challenge. This study introduces the Hydrological Attribution and Analysis Framework (HAAF), a novel three-stage methodology, to provide a comprehensive explanation for the nearly-centennial (1931-2024) water balance dynamics of Lake Balkhash. The HAAF first establishes a high-fidelity hydrological reconstruction using a Physics-Informed Machine Learning (PIML) model, then employs the Budyko framework to attribute runoff changes, and finally links these catchment-scale drivers to the lake's terminal water balance. Our results confirm the robustness of the PIML model in simulating historical runoff (KGE > 0.75). The attribution analysis then reveals a complex interplay of competing forces. During the intensive intervention period (1970-1990), a substantial human-induced runoff reduction of -9.21 km<sup>3</sup> completely masked a significant climate-driven wetting potential (+6.13 km<sup>3</sup>), triggering the lake's sharp decline. In the recent period (1991-2024), the basin's hydrology has been governed by a fragile stalemate in which a massive, climate-driven potential for increased runoff (+10.80 km<sup>3</sup>) was almost entirely neutralized by the persistent negative impact of human water use (-11.36 km<sup>3</sup>). At the lake level, this translated into an apparent stability sustained only by a favorable climatic subsidy. Future projections under various climate scenarios indicate that this climatic buffer is transient and unlikely to persist, exposing the lake to a high risk of rapid decline. We conclude that the recent stability of Lake Balkhash is not a sign of systemic recovery but a "masked vulnerability." This highlights the urgent need for sustainable and forward-looking water management strategies that account for these underlying, competing drivers.

## 1 Introduction


Endorheic lakes in arid and semi-arid regions are widely recognized as sensitive indicators of hydroclimatic change (Zhang et al., 2021). The water level and ecological health of these lakes are governed by a delicate balance between water inputs from their catchments and evaporative losses. Today, this balance is increasingly under pressure from two primary forces: global climate change, which alters precipitation patterns, temperature, and cryospheric contributions, and direct human activities,




such as water withdrawal for agriculture and reservoir regulation (Immerzeel and Bierkens, 2012; Li et al., 2016; Mandal and Chanda, 2023). For instance, the Aral Sea has dramatically shrunk due to irrigation diversions, while Lake Urmia faces similar threats from water abstraction and drought. These examples underscore the global relevance of understanding water balance dynamics in closed basins, where water cycles are tightly coupled with climate and anthropogenic activities. Disentangling the individual impacts of these concurrent drivers is a fundamental challenge in hydrology and Earth system science. A robust quantitative attribution is not only crucial for understanding past hydrological dynamics but also essential for developing sustainable water management strategies and predicting the future trajectory of these vital ecosystems.

Lake Balkhash exemplifies these interactions, with its large basin situated in southeastern Kazakhstan, primarily fed by the Ili River which originates in the Tianshan Mountains (Duan et al., 2020). Historical fluctuations include sharp declines in the 20th century, followed by relative stability, amid accelerating warming and human water use (Duan et al., 2021). As an endorheic lake, Lake Balkhash has no outlet, and all incoming runoff is ultimately lost to evaporation. Its primary source region is experiencing a warming rate significantly higher than the global average, accelerating glacial melt (Jin et al., 2024). While this melt initially boosts lake inflow in the short term, it signals a long-term depletion of solid water reserves. This shifting composition of water sources heightens the lake's sensitivity to climatic fluctuations, underscoring the profound impact of climate change on its hydrology. Concurrently, anthropogenic activities, particularly water consumption in the lower Ili River basin, have been identified as a key driver of the lake's water level decline in the latter half of the 20th century, sparking significant concerns regarding regional water management and sustainability (Cai et al., 2014).

Reduced inflow not only directly lowers the water level of Lake Balkhash but also triggers a cascade of ecological consequences, such as delta ecosystem degradation (Starodubtsev and Truskavetskiy, 2011), increased lake salinity (Shen et al., 2021), and the destruction of aquatic habitats (Li et al., 2021). Although previous research has identified climate and human activities as the primary drivers of runoff change and attempted to quantify their relative contributions (Gan et al., 2022; Wang et al., 2024; Yu et al., 2025), these studies often fail to integrate their findings with the lake's terminal water balance, hindering a comprehensive explanation. The region's scarcity of long-term, high-quality observational data on inflow, especially for naturalized flow (i.e., runoff unaltered by human interference), poses a significant challenge. For the same reason, simulations using traditional hydrological models or machine learning (ML) methods in this area are subject to high uncertainty. These limitations impede a clear differentiation and comparison of the impacts of climate change and human activities across different periods, leaving a critical knowledge gap regarding the dominant factors governing the lake's water fluctuations.

To address these existing gaps, this study introduces and implements the Hydrological Attribution and Analysis Framework (HAAF), a methodology designed to rigorously distinguish the drivers of hydrological change. The strength of HAAF lies in its structured approach, which first employs a PIML model for high-fidelity hydrological reconstruction to generate robust long-term interannual runoff time series. It then utilizes the Budyko framework for quantitative driver attribution, separating climatic from anthropogenic impacts on runoff. Finally, the framework provides a system-level impact linkage by connecting these attributed catchment-scale changes directly to the lake's water balance dynamics. This integrated workflow allows for a


complete diagnosis from cause to effect, thereby offering a perspective on studying the water balance of lakes in arid regions, comparable to studies on other lakes where similar attribution challenges persist.

The main objective of this paper is to provide a comprehensive quantitative explanation for the centennial dynamics of Lake Balkhash's water volume by applying the HAAF methodology. To achieve this, our study is structured around the three core stages of the HAAF framework:

- (1) Reconstructing the annual naturalized and human-impacted runoff into Lake Balkhash for the past century (1931-2024), employing the PIML model as the high-fidelity reconstruction engine.
- (2) Quantitatively separating the relative contributions of climate change and direct human activities to the observed runoff changes across three distinct historical periods, using the Budyko framework for attribution.
- (3) Elucidating how these attributed changes in water inputs have governed the historical water storage fluctuations of Lake Balkhash by linking them through a lake water balance equation.

### 2 Materials and Methods

## 2.1 Study Area and Historical Periodization

Figure 1: Geographic location of the study area.

Lake Balkhash, one of the largest endorheic lakes in the arid region of Central Asia, is situated in southeastern Kazakhstan (Duan et al., 2020). It covers an area of approximately 16,400 km², has an average depth of 5.8 meters, and exhibits pronounced differences in water quality between its western (freshwater) and eastern (saline) sections (Shen et al., 2021). The lake receives inflow from five primary rivers, predominantly the Ili River, which accounts for over 70% of the total inflow (Liu et al., 2024).

Other significant tributaries include the Karatal, Aksu, Lepsy, and Ayaguz rivers, originating from the Tien Shan and Dzungarian Alatau mountains. The Lake Balkhash basin encompasses a total area of approximately 413,000 km² and features an elevation range from 60 to 6,000 m (Fig. 1). The mountainous upper reaches of the river are an area of widespread glaciers and seasonal snowpack, and these ice/snowmelt waters have traditionally been one of the main sources of water for Lake Balkhash. The entire basin is deep within the Eurasian continent, with a typical temperate continental arid climate, sparse and uneven spatial distribution of precipitation - the mountainous areas receive significantly more precipitation than the plains and lakes (Cao et al., 2022).

The hydrological history of Lake Balkhash is marked by critical turning points that have reshaped its ecosystem, influenced by both natural variability and human activities. During the 1970s and 1980s, the lake underwent a severe ecological crisis, principally driven by the impoundment of the Kapchagay Reservoir on its main tributary, the Ili River (Yu et al., 2025). This event triggered a sharp decline in the lake's water level, a corresponding rise in salinity, and significant biodiversity loss. Following the dissolution of the Soviet Union in 1991, while direct reservoir impoundment pressures lessened, subsequent political and economic shifts introduced new complexities, intensifying challenges in regional water allocation and altering regional water management paradigms (Jia et al., 2020). In light of these pivotal events, we have structured our investigation by dividing the entire study period (1931-2024) into three distinct periods based on the 1970 and 1991 milestones to facilitate a dynamic attribution analysis. This segmentation aligns with previous studies that have validated its reasonableness for capturing shifts in hydrological drivers (Wang et al., 2024). A summary of these periods is presented in Table 1.

Table 1 Summary of the three defined historical periods

| Period | Time Frame | Designation                   | Key Characteristics                                                                                                                                |  |
|--------|------------|-------------------------------|----------------------------------------------------------------------------------------------------------------------------------------------------|--|
| P1     | 1931-1969  | Reference Period              | Limited direct hydrological intervention, with water availability primarily governed by natural climate variability.                               |  |
| P2     | 1970-1990  | Intensive Intervention Period | Dominated by intense human interference, where major<br>hydrological engineering fundamentally altered the<br>regional water balance.              |  |
| Р3     | 1991-2024  | Compounded Pressures Period   | Defined by the combined effects of stabilized engineering impacts, post-Soviet shifts in water management policy, and accelerating climate change. |  |

#### 2.2 Datasets



This study employed a diverse range of datasets to support the hydrological modeling and analysis of hydro-climatic changes. These datasets include a Digital Elevation Model (DEM), soil properties, land use/land cover (LULC) maps, glacier inventories, glacier elevation changes, meteorological forcings, and observed streamflow data. A summary of these datasets is presented in Table 2, including key variables, spatial and temporal resolutions, coverage periods, and access links for reproducibility.

Table 2: Summary of datasets used in this study




| Dataset                                        | Key Variables                                                               | Spatial<br>Resolution          | Temporal Coverage      | Source                                    |
|------------------------------------------------|-----------------------------------------------------------------------------|--------------------------------|------------------------|-------------------------------------------|
| Copernicus<br>GLO-90 DEM                       |                                                                             | 90 m                           | Static                 | OpenTopography                            |
| DSOLMap                                        | Bulk density, hydraulic conductivity, available water capacity              | 250 m                          | Static                 | WateriTech                                |
| GLC_FCS30E                                     | Land cover classes (35 subcategories)                                       | 30 m                           | 1985–2022              | Zenodo                                    |
| Randolph<br>Glacier<br>Inventory<br>(RGI v7.0) | Glacier outlines, attributes                                                | Vector                         | ~2000 snapshot         | GLIMS                                     |
| SWORD v15                                      | River reaches, nodes, hydrological networks, lake boundaries                | ~10 km reaches,<br>200 m nodes | Static                 | Zenodo                                    |
| Hugonnet e al. (2021)                          | t Glacier elevation change rates (dh/dt)                                    | 100 m                          | 2000–2019              | Original publication                      |
| CRU JRA v2.5                                   | vapor pressure, etc.                                                        | to 0.05°)                      | 1901–2024 (daily)      | CEDA Archive                              |
| TerraClimate                                   | Max/min temperature, precipitation, solar radiation, vapor pressure deficit | 1/24°                          | 1958–present (monthly) | Climatology Lab                           |
| Observed<br>Streamflow                         | Discharge (daily/monthly)                                                   | Point                          | Varies (1931–2024)     | National Cryosphere<br>Desert Data Center |

The Digital Elevation Model used was the Copernicus GLO-90 DEM, which provides elevation data at a spatial resolution of 90 meters and is accessible via OpenTopography. Soil hydraulic parameters (e.g., bulk density, hydraulic conductivity, available water capacity) were derived from DSOLMap (Lopez-Ballesteros et al., 2023), a 250m-resolution dataset available through WateriTech. For Land Use/Land Cover, we used the GLC FCS30D dataset (Zhang et al., 2024), which offers the highest available resolution (30m) for a global, long-term LULC time series covering 1985-2022 with 35 land-cover subcategories, and is available on the Zenodo platform. Glacier outlines and attributes were obtained from the Randolph Glacier Inventory v7.0 (RGI), provided by the Global Land Ice Measurements from Space (GLIMS) initiative. Hydrological networks and lake boundaries were delineated using the Surface Water and Ocean Topography mission river database (SWORD) v15, which is based on a 30m DEM and has demonstrated superior accuracy compared to the widely used HydroSHEDS dataset (Altenau et al., 2021). Glacier elevation change data from the study by Hugonnet et al. (2021) were used for calibrating glacier melt parameters. For historical climate forcing data, we primarily utilized the CRU JRA v2.5 dataset (an update to the originally referenced v3.0 for consistency with recent releases), which provides daily meteorological variables (e.g., temperature, precipitation, wind speed, vapor pressure) at a 0.5° spatial resolution from 1901 to 2024, making it the longest daily historical climate forcing dataset currently available. It is accessible through the CEDA Archive. This dataset is derived from the CRU TS dataset, whose reliability in Central Asia after 1930 has been confirmed in numerous studies (Duan et al., 2020). Based on this confirmed post-1930 reliability, our study period was set from 1931 to 2024 to ensure the accuracy of our simulations. To enhance the precision of these climate inputs, particularly for smaller sub-basins, the CRU JRA data was downscaled to a finer 0.05° resolution. This was achieved using the delta-change statistical downscaling method with bias

correction (Peng et al., 2019), leveraging the monthly TerraClimate dataset (1/24° resolution), which is also derived from CRU TS, to ensure consistency across datasets. The observed streamflow dataset is provided by the National Cryosphere Desert Data Center and previous related studies (Duan et al., 2020; Guo et al., 2015).

#### 2.3 Methodology

The core methodology of this study is the Hydrological Attribution and Analysis Framework (HAAF), a structured three-stage process designed to attribute the causes of lake water volume changes. The workflow begins with hydrological process modeling in the catchment, followed by driver attribution, and concludes by linking these drivers to the lake's response. Figure 2 shows a conceptual flowchart, and the following is a detailed description of the three stages.

Figure 2: AAF Flowchart (Step 1: PIML-based Reconstruction, Step 2: Budyko Framework Attribution, Step 3: Lake System Response Linkage, with inputs like dynamic forcings and outputs like ΔV<sub>climate</sub> and ΔV<sub>human</sub>.)

## 2.3.1 PIML-based Hydrological Reconstruction


Physics-informed machine learning (PIML) integrates conceptual hydrological models with machine learning (ML) to synergistically leverage the predictive power of ML algorithms and the process-based understanding of physical models (Bhasme et al., 2022). This integration also enhances the model's physical consistency and interpretability. The SWAT model and its improved versions are widely used in hydrological simulation processes. In our framework, we employed the Sublimation-Enhanced Glacier SWAT+ model (SEGSWAT+), which is an iteration of the SWAT model specifically adapted





for glacial hydrology and has been proven effective for simulating the glacially-influenced runoff in the Lake Balkhash basin (Yang et al., 2024). By combining inputs (precipitation, potential evapotranspiration), state variables (groundwater storage, soil moisture), and intermediate outputs (actual evapotranspiration) leading to a target variable (simulated runoff at specific gauge locations). Subsequently, a suite of ML methods competed to learn the residuals—the discrepancies between SEGSWAT+ simulated runoff and observed streamflow—to correct and improve the final output accuracy. The architecture of our PIML framework is illustrated in Fig. 3. The rationale for this PIML design is that the residual between a model prediction and an observation represents a composite error stemming from inherent limitations of the process-based hydrological model and uncertainties in driving data. By learning these complex residual patterns, the ML component enhances hydrological accuracy without severe overfitting biases, as residuals from a reliable physical model are inherently bounded and cannot exceed the runoff magnitude, constraining the ML model and preserving physical plausibility.

For the ML component, we employed a diverse ensemble of architectures common in hydrology and Earth science applications, including Artificial Neural Networks (ANN), Long Short-Term Memory (LSTM) networks, Random Forest (RF), and XGBoost (Behrouz et al., 2022; Guo et al., 2023; Srinivasulu and Jain, 2006; Wang and Peng, 2024). The best-performing model from this competitive ensemble was selected for each simulation period, a method known for its robustness in capturing water balance dynamics. The proposed PIML model was run for the three distinct periods, calibrated using multi-objective functions (including Kling-Gupta Efficiency (KGE), Nash-Sutcliffe Efficiency (NSE), and percent bias) against data from 16 hydrological stations, with cross-validation to ensure generalizability.

Figure 3 PIML structure

## 2.3.2 Budyko-based Driver Attribution

To quantitatively distinguish the impacts of climate change from those of direct human activities on runoff, we employed the Budyko framework (Budyko and Miller, 1974). This framework provides a robust, first-order approximation of the long-term water balance in a catchment by describing the partitioning of precipitation (P) into actual evapotranspiration (E) and runoff (Q). Its core hypothesis is that the ratio of actual evapotranspiration to precipitation (E/P) is primarily a function of the aridity index  $(\Phi)$ , defined as the ratio of potential evapotranspiration  $(E_0)$  to precipitation (P). The total change in mean annual runoff  $(\Delta Q)$  between a baseline and an altered period can be decomposed into contributions from climate change  $(\Delta Q_c)$  and direct


human activities ( $\Delta Q_h$ ). For this decomposition, we adopted a climate elasticity method (Dooge et al., 1999). This method approximates the contribution of any controlling factor x (e.g., the precipitation P) to the total runoff change ( $\Delta Q$ ) as the product of the change in that factor ( $\Delta x$ ) and the sensitivity of runoff to that factor ( $\partial Q/\partial x$ ). This relationship is mathematically grounded in the widely used Choudhury-Yang equation (Yang et al., 2008), i.e., Equation 1, which is a one-parameter form of the Budyko curve:

$$\frac{E}{P} = \frac{1}{[1 + (P/E_0)^n]^{1/n}} \tag{1}$$

This equation features a single catchment parameter, n, which represents the integrated control of the underlying landscape characteristics (e.g., vegetation, soil, topography) on the partitioning of effective precipitation into runoff and evapotranspiration. Consequently, the change in this parameter ( $\Delta n$ ) between periods can be interpreted as the integrated effect of direct human activities that alter these landscape properties (e.g., land use change, reservoir construction).

Following this framework, the total runoff change ( $\Delta Q$ ) can be partitioned into contributions from changes in climatic variables (rainfall, snowmelt, glacial melt,  $E_0$ ) and human-induced landscape changes (represented by  $\Delta n$ ). The sensitivity coefficients of runoff to each of these factors are detailed in Equations (2a-2c), and equation (3) provides a comprehensive expression:

$$\frac{\partial ET}{\partial P} = \frac{ET}{P} \left( \frac{ET_0^n}{ET_1^n + P^n} \right) \tag{2a}$$

$$\frac{\partial ET}{\partial ET_0} = \frac{ET}{ET_0} \left( \frac{P^n}{ET_0^n + P^n} \right) \tag{2b}$$

$$\frac{\partial ET}{\partial n} = \frac{ET}{n} \left( \frac{\ln(ET_0^n + P^n)}{n} - \frac{ET_0^n \ln ET_0 + P^n \ln P}{ET_0^n + P^n} \right) \tag{2c}$$

$$\Delta Q = \left(1 - \frac{\partial ET}{\partial P}\right) \Delta P - \frac{\partial ET}{\partial ET_0} \Delta ET_0 - \frac{\partial ET}{\partial n} \Delta n \tag{3}$$

## 185 2.3.3 Lake System Response Linkage

The change in Lake Balkhash's water storage ( $\Delta V$ ) over a defined period ( $\Delta t$ ) is essentially determined by the balance between water inflow and water dissipation. Its water balance equation can be expressed as follows:

$$\frac{\Delta V}{\Delta t} = A(h)(P - E) + Q_{in} \tag{4}$$

Where A is the lake surface area, which is a function of water level (h),  $Q_{in}$  is the inflow. The time series of annual water levels for Lake Balkhash during the study period was obtained from the work of Nakayama et al. (1997) and Duan et al. (2020) Subsequently, the corresponding lake surface area time series was derived using the level-to-area conversion function provided by the former. The  $\Delta V$  was then calculated based on the methodology established by Zhang et al. (2013), an approach whose reliability for Lake Balkhash has been previously validated (Wang et al., 2022). It is important to note that, given the long-

205

term temporal scale of this study, the exchange between the lake and surrounding groundwater systems was considered negligible and thus omitted from the water balance calculations.

The final stage of the HAAF connects the attribution results from Stage 2 to the observed changes in the lake's water storage, thereby achieving an end-to-end attribution. This linkage is achieved through the lake's water balance equation. The total change in lake storage ( $\Delta V_{obs}$ ) between two periods (e.g., an altered period vs. a baseline period) can therefore be expressed as:

$$200 \quad \Delta V_{obs} = \Delta Q_{in} + \Delta P - \Delta E \tag{5}$$

The key innovation in Stage 3 is to decompose this total storage change into contributions from climate change and human activities. We achieve this by substituting the attributed runoff changes from Stage 2 into Equation (5). The  $\Delta Q_{in}$  was already separated into  $\Delta Q_c$  and  $\Delta Q_h$  components. Changes in lake precipitation and lake evaporation are, by definition, driven by climatic factors. Therefore, we can re-organize Equation (5) to separate the total storage change  $(\Delta V_{obs})$  into its climatic  $(\Delta V_c)$  and human-activity  $(\Delta V_h)$  driven components:

$$\Delta V_C = \Delta Q_C + \Delta P - \Delta E \tag{6a}$$

$$\Delta V_C = \Delta Q_C + \Delta P - \Delta E \tag{6b}$$

This set of equations (6a and 6b) represents the core of the "attribution transference" in the HAAF framework. It allows us to quantitatively determine how much of the observed change in the lake's total water volume is due to natural climate variability (acting on both the catchment and the lake itself) and how much is due to direct human activities. This provides a complete, system-level quantitative explanation for the lake's historical dynamics.

#### 2.3.4 Model Evaluation and Uncertainty Metrics

The hydrological simulation performance of the PIML model was rigorously evaluated by comparing the simulated daily and monthly streamflow against observed data. For this purpose, we selected three widely used metrics: the Coefficient of Determination (R2), the Kling-Gupta Efficiency (KGE), Percent Bias (PBIAS), and the Nash-Sutcliffe Efficiency of the logarithm of streamflow (logNSE) (Yang et al., 2023). To evaluate the agreement between simulated and observed values, the R2 was calculated, which measures the proportion of the variance in the dependent variable that is predictable from the independent variables. KGE is a comprehensive metric that decomposes model performance into three distinct components: correlation (r), bias ratio (β), and variability ratio (γ). This decomposition allows for a more nuanced diagnosis of model deficiencies by distinguishing between errors in timing (correlation), overall water balance (bias), and flow dynamics (variability). KGE is calculated as follows:

$$KGE = 1 - \sqrt{(r-1)^2 + (\beta - 1)^2 + (\gamma - 1)^2}$$
(7)


where  $\beta = \mu_s/\mu_o$  and  $\gamma = (\frac{\sigma_s}{\mu_s})/(\frac{\sigma_o}{\mu_o})$ .  $\mu$  and  $\sigma$  represent the mean and standard deviation of the simulated (s) and observed (o) streamflow, respectively, and r is the Pearson correlation coefficient between them. A KGE value greater than -0.41 is generally considered acceptable for hydrological modeling, indicating that the model performs better than the mean flow benchmark. PBIAS measures the average tendency of the simulated data to be larger or smaller than their observed counterparts, providing a straightforward assessment of model bias. It is calculated as:

$$PBIAS = \sum_{i=1}^{n} \frac{Q_{sim} - Q_{obs}}{Q_{obs}} \times 100 \tag{8}$$

where  $Q_{sim}$  represents the simulated runoff and  $Q_{obs}$  represents the observed runoff. A PBIAS value within the range of  $\pm 25\%$  is typically considered satisfactory, with values closer to 0 indicating better model performance. The log~NSE is a modification of the standard Nash-Sutcliffe Efficiency (NSE) that is particularly sensitive to model performance during low-flow periods. This is critically important for hydrological modeling in arid and semi-arid regions. It is calculated as:

$$\log NSE = 1 - \frac{\sum_{i=1}^{n} (\log Q_{sim} - \log Q_{obs})^2}{\sum_{i=1}^{n} (\log Q_{obs} - \overline{\log Q_{obs}})^2}$$
(9)

where  $\overline{\log Q_{obs}}$  represents the mean of the logarithm of the runoff observations. A log~NSE value greater than 0.36 is generally considered to indicate a satisfactory to good model performance for low-flow simulations.

#### 3. Results




## 3.1 Hydrological Model Performance Evaluation

Accurate hydrological runoff modeling is the first step in HAAF. Based on the PIML model, parameterization is performed by combining field measurements, geospatial datasets, and remote sensing products (e.g., topographic data, soil texture maps, and soil hydraulic parameters). The calibrated parameters included: (1) snow module parameters (critical melt temperature and degree-day factors); (2) vegetation and land surface parameters (root depth, soil anisotropy ratio, surface depression storage capacity, and surface roughness); and (3) glacier module parameters (critical melt temperature, degree-day factors, and areavolume scaling parameters. Detailed calibration parameters can be referenced in the study by Yang et al (2022). Model calibration and validation were performed independently for each of the three study periods, resulting in three distinct parameter sets. Notably, the parameter set for P1 was specifically configured to simulate naturalized streamflow under conditions free of significant human intervention. For each period and station, the available observed streamflow data were partitioned into calibration (70%) and validation (30%) subsets.

The predictive performance of machine learning models was evaluated using the R2 metric to determine the optimal model for modeling the residual runoff (Fig. 4). While the Random Forest (RF), Long Short-Term Memory (LSTM), and XGBoost models all demonstrated high predictive accuracy, with respective R2 values of 0.865, 0.863, and 0.912, XGBoost distinguished itself. Its predictions not only achieved the highest coefficient of determination but were also the most tightly



clustered around the 1:1 reference line, indicating superior stability and minimal variance. In contrast, the Artificial Neural Network (ANN) exhibited inferior performance (R2 = 0.810), characterized by a widely scattered data distribution and significant deviation from the 1:1 line, which suggests substantial bias. Given its superior accuracy and reliability, XGBoost was therefore selected as the optimal model to adjust the final streamflow simulations.

Figure 4 Scatter plot of residual predictions and actual values

Based on the optimal machine learning model, the final runoff results obtained were compared with the measured runoff data through a multi-indicator assessment. The comprehensive performance assessment results are shown in Fig. 5. The first ten hydrological stations, which control the main stem of the Ili River, demonstrated strong model performance. For all these stations, KGE values exceeded 0.75, PBIAS was within  $\pm 10\%$ , and logNSE was above 0.7, indicating an excellent fit to the observed runoff. Specifically, the Ushzharma station, located at the river's terminus just before it enters the lake, exhibited outstanding performance with a KGE of 0.847, a low bias (PBIAS = 5.1%), and exceptional skill in simulating low flows (logNSE = 0.929). The remaining stations, which monitor the four eastern tributary rivers, showed similarly robust results. KGE values were consistently above 0.8 and PBIAS remained within  $\pm 10\%$ . Except for the Chiganak station, where the logNSE of 0.699 is still considered indicative of good low-flow simulation, all other eastern stations achieved logNSE values greater than 0.7. This confirms that the model accurately simulates runoff for all major rivers flowing into the eastern part of

the lake. This robust performance establishes a solid foundation for the subsequent lake water balance calculations and the application of the Budyko framework for attribution analysis.

Figure 5 Results of the runoff fitting assessment




## 3.2 Quantification of The Impacts on Variations in Runoff

The total runoff into Lake Balkhash was calculated as the sum of simulated streamflow from all modeled tributary rivers, adjusted for deltaic water consumption using the methodology of Thevs et al. (2016). The simulation strategy to generate the final time series was as follows: Naturalized Runoff (Qnat), the parameter set calibrated for the baseline period (P1) was used to simulate naturalized streamflow across the entire study period (1931-2024). Human-Impacted Runoff (Qreal), the construction of the historical, human-impacted runoff series involved a multi-step process. The parameter set for P2 was used to simulate runoff for both the P1 and P2 periods (1931-1990), while the P3 parameter set was applied exclusively to its own period (1991-2024). For any year where direct observational data were available, these records superseded the simulated values to create the final composite Qreal series. The results of the reconstruction of interannual natural runoff and actual runoff over nearly a century are shown in Fig. 6.




Figure 6 Natural runoff and actual runoff into the lake

During the baseline period (1931-1969), the Qnat and Qreal time series exhibit a remarkably close correspondence, tracking each other's inter-annual fluctuations with high fidelity. Analysis of the deviations between naturalized (Qnat) and observed (Qreal) runoff reveals a distinct three-stage evolution of human impact on the basin's hydrology. During the baseline period (P1, 1931-1969), the runoff deficit was minimal, with a mean deviation of only 0.48 km<sup>3</sup>/yr. The small range between the maximum (1.29 km<sup>3</sup>/yr) and minimum (-0.19 km<sup>3</sup>/yr) differences indicates that the observed flow was closely aligned with natural conditions, fluctuating slightly around the naturalized values. In stark contrast, the intensive intervention period (P2, 1970-1990) marked a dramatic and persistent shift. The mean runoff deficit surged to 3.11 km<sup>3</sup>/yr, and critically, the minimum deviation remained positive (0.93 km<sup>3</sup>/yr), signifying a systematic and sustained reduction in flow throughout every year of this period, directly attributable to the onset of large-scale human activities. Following this, the compounded stress period (P3, 1991-2024) exhibited a more complex pattern. The mean runoff deficit decreased to 1.35 km<sup>3</sup>/yr, substantially lower than in P2, suggesting a partial mitigation or a shift in human impact, possibly due to changes in water management policies. Overall, the statistical progression clearly quantifies the transition from a near-natural state to a period of intense, sustained water withdrawal, and finally to a recent era characterized by reduced average impact but greater variability and more extreme events. The Budyko framework was employed to diagnose the hydro-climatic shifts between the three periods and to quantitatively attribute the changes in streamflow. The basin's evolutionary trajectory in the Budyko space is shown in Fig. 7 (a). The Budyko analysis, based on a calibrated catchment parameter (n) of 1.776, quantitatively confirms the dominant role of human activities in altering the basin's hydrology, particularly during the period of intensive intervention. From the baseline period (P1) to the period of intensive alteration (P2), the total observed runoff decreased by 0.31 km<sup>3</sup>/yr. Our attribution results are unequivocal: direct human activities were the overwhelming driver of this decline, accounting for -0.27 km<sup>3</sup>/yr, or 86.3% of the total change. Remarkably, this occurred while the climate became slightly less arid (aridity index  $\phi$  decreased from 1.404 to 1.349). This finding underscores that the significant hydrological deficit during this era was not a consequence of adverse climate conditions

325

but was almost entirely driven by anthropogenic water withdrawals, consistent with the reality of large-scale water storage at the Kapchagay Reservoir.

Figure 7 Budyko analysis and attribution of runoff changes. (a) Trajectory of the basin's hydro-climatic conditions across three periods. (b) Attribution of total runoff changes between periods to climate and human activity contributions

In the subsequent period (P3), the dynamic between drivers shifted dramatically. The observed runoff nearly recovered to baseline levels (a minor change of -0.06 km³/yr), but this apparent stability masks a critical dynamic of competing forces. The climate trend, on its own, would have led to a modest increase in runoff (+0.02 km³/yr) due to a continued shift toward less arid conditions (Φ = 1.287). However, this potential climatic gain was entirely offset and surpassed by the persistent negative impact of human activities (-0.08 km³/yr). This analysis highlights a crucial transition: while the absolute magnitude of human impact lessened compared to P2, it remained strong enough to counteract a favorable climate trend, preventing a full hydrological recovery. The system has thus evolved from one dominated by direct human intervention to one where human water use actively suppresses the benefits of a wetter climate cycle.

A detailed attribution analysis, which decomposes the climatic contribution into its constituent parts, reveals a powerful but hidden dynamic of competing forces that have fundamentally reshaped the basin's hydrology (Table 2). During the period of intensive alteration (P2 vs. P1), the climate, on its own, would have caused a massive increase in runoff, calculated at +6.13 km³. This seemingly paradoxical result was driven by a very large decrease in potential evapotranspiration (PET), which contributed +19.50 km³ to runoff, overwhelmingly compensating for the severe decline in glacial melt (-13.71 km³). However, this substantial climatic gain was entirely erased by the immense impact of direct human activities, which caused a -9.21 km³ reduction in flow. This demonstrates that the observed moderate decline in runoff was the net result of a massive climate-driven water surplus being entirely offset by equally massive anthropogenic withdrawals.

Table 3 Decomposition of Climate and Human Impacts on Runoff

|           | P2 vs P1     |                         |              |                             |
|-----------|--------------|-------------------------|--------------|-----------------------------|
| Component | $\Delta x_i$ | $\partial Q/\partial x$ | $\Delta Q_i$ | $\Delta Q_{i}/\Delta Q(\%)$ |



| Rain          | 0.674        | 1.335                   | 0.899                   | -29.2                     |
|---------------|--------------|-------------------------|-------------------------|---------------------------|
| $Melt_{snow}$ | -0.421       | 1.335                   | -0.562                  | 18.3                      |
| Meltglacier   | -10.270      | 1.335                   | -13.709                 | 445.4                     |
| PET           | -18.153      | -1.074                  | 19.500                  | -633.5                    |
| n             | /            | /                       | -9.206                  | 299.1                     |
|               | P3 vs P1     |                         |                         |                           |
| Component     | $\Delta x_i$ | $\partial Q/\partial x$ | $\Delta Q_{\mathrm{i}}$ | $\Delta Q_i/\Delta Q(\%)$ |
| Rain          | 7.801        | 1.335                   | 10.414                  | -187.5                    |
| $Melt_{snow}$ | -0.264       | 1.335                   | -0.352                  | 63.4                      |
| Meltglacier   | -12.468      | 1.335                   | -16.643                 | 299.7                     |
| PET           | -16.183      | -1.074                  | 17.384                  | -313.0                    |
| n             | /            | /                       | -11.358                 | 204.52                    |

This pattern of opposing forces intensified dramatically in the most recent period (P3 vs. P1). The climate-driven potential for runoff generation soared to +10.80 km³, again fueled primarily by a large decrease in PET (+17.38 km³) and a significant increase in rainfall (+10.41 km³). These gains were more than enough to offset the continued and worsening decline in glacial melt (-16.64 km³). Yet, despite this enormous climatic potential for a wetter regime, the observed runoff remained near baseline levels. This was because the negative impact of human activities also intensified to -11.36 km³, effectively neutralizing the entire climate-driven surplus. These findings expose a critical vulnerability: the basin's apparent hydrological stability is a fragile illusion, maintained only by a coincidental and likely temporary opposition of very large, opposing forces. Any change in this delicate balance—such as a return of PET to higher levels or a decrease in rainfall—could expose the system to the full, severe impact of its lost cryospheric storage and sustained human water demand.




## 3.3 Lake System Response to Changes in Water Supply

Figure 8 simulated water volume changes and water volume changes calculated based on lake data

volume changes ( $\Delta V$ ) with the simulated volume changes ( $\Delta V$ sim). The reconstructed  $\Delta V$  was derived from satellite-based water level and area data, representing the observed reality of the lake's storage change. The simulated  $\Delta V_{sim}$  was calculated independently as the net result of our modeled total inflow minus lake surface evaporation. As shown in Fig. 8, there is a strong visual correspondence between the two time series, with the model successfully capturing the magnitude and timing of the major inter-annual fluctuations. This visual agreement is substantiated by robust statistical metrics: the correlation coefficient (R) is 0.86 (p < 0.01), indicating a highly significant positive relationship, and the mean bias is exceptionally low at 0.135 km<sup>3</sup>/yr. This strong agreement serves as a crucial cross-validation, confirming that our framework accurately closes the water balance at the lake terminus. Since the simulated volume change is directly dependent on the accuracy of our reconstructed inflow, this result provides high confidence in the reliability of the runoff data used for the subsequent attribution analysis. The dynamic reconstruction of the Lake Balkhash water balance, illustrated in Fig. 9, reveals a dramatic history of change driven primarily by fluctuations in river inflow. During P1, the lake existed in a state of relative equilibrium. Mean annual inflow of 18.4 km<sup>3</sup>/yr was nearly balanced by lake surface evaporation of 20.4 km<sup>3</sup>/yr, resulting in a negligible mean storage change ( $\triangle$ V) of +0.07 km<sup>3</sup>/yr. The lake experienced years of water gain and loss in roughly equal measures (46% vs. 54%). This stability was abruptly shattered in the P2. A sharp decline in mean annual inflow to 15.3 km<sup>3</sup>/yr, a drop of over 3 km<sup>3</sup>/yr, directly tipping the balance into a state of sustained deficit. Despite a concurrent decrease in evaporation, the reduced input was so significant that the lake's storage declined by an average of 1.0 km<sup>3</sup>/yr, with two-thirds of the years in this period experiencing a net water loss and a cumulative volume reduction of 21.0 km<sup>3</sup>.

To provide a final, integrated validation of our entire modeling framework, we compared the reconstructed annual lake water




Figure 9 Long-term water balance of Lake Balkhash

The P3 marked a phase of partial recovery and re-stabilization. Inflow rebounded significantly to 17.8 km<sup>3</sup>/yr, approaching pre-1970 levels. This recovery in water supply was sufficient to counteract the evaporative losses (19.8 km<sup>3</sup>/yr), bringing the mean storage change back to a near-neutral state (+0.04 km<sup>3</sup>/yr) and restoring a majority of years (55.9%) to a positive water balance. A noteworthy and perhaps counter-intuitive finding is the statistically significant, albeit slight, decreasing trend in total lake surface evaporation over the entire 1931-2024 period (p < 0.05). This suggests that despite a warming climate, other factors such as changes in wind speed, humidity, or lake surface thermal dynamics may have modulated and even slightly reduced long-term open-water evaporation. Across all periods, river inflow consistently accounted for ~90% of the total water input, making it the primary lever controlling the lake's state. The lake's storage volume, and by extension its water level, is thus extremely sensitive to the marginal difference between these massive inputs and outputs. The ~17% reduction in inflow during P2 was sufficient to trigger a prolonged period of decline, while its subsequent recovery was the key factor in the lake's recent stabilization. This highlights the critical dependence of this vital ecosystem on the hydrological health of its catchment. To synthesize our findings into a comprehensive explanation for the lake's historical trajectory, we performed a final attribution analysis on the changes in the mean annual lake water storage, with the results presented in Fig. 10. During the intensive alteration period (P2 vs. P1), the lake's water balance paradoxically improved, with the mean storage change increasing by a substantial +4.54 km<sup>3</sup>. This was driven by a powerful, positive climatic contribution of +7.17 km<sup>3</sup>, primarily caused by a significant reduction in lake surface evaporation. However, this massive climatic benefit was simultaneously counteracted by the negative impact of direct human activities (-2.63 km<sup>3</sup>), which, through reduced inflow, consumed more than a third of the


potential gain. This reveals that the lake's water balance improved despite, not because of, the anthropogenic pressures of that era.

This dynamic of opposing forces became even more pronounced in the most recent period (P3 vs. P1). The positive climatic contribution soared to an immense +12.49 km³, fueled by both a continued decrease in evaporation and a substantial increase in direct precipitation over the lake. While the negative impact of human activities persisted (-0.87 km³), it was dwarfed by the magnitude of the favorable climate trend, resulting in a large net increase in the lake's storage change of +11.62 km³. Ultimately, these results provide a critical insight: the apparent hydrological stability and recovery of Lake Balkhash are not a sign of intrinsic resilience but are instead precariously balanced and heavily subsidized by a highly favorable, and potentially transient, climatic trend. The persistent underlying water deficit caused by human use remains, masked by this climatic boon. Any reversal of these favorable climate conditions could rapidly expose the lake's profound vulnerability.

Figure 10 Attribution of changes in Lake Balkhash water Storage

## 4. Discussion



This study successfully introduced and implemented a novel attribution framework to disentangle the complex, century-long interplay between climate change and human activities on the hydrology of the Lake Balkhash basin. Our results not only quantifying the drivers of streamflow change but also by explicitly linking these changes to the lake's ultimate water balance response. A key finding is that the basin's hydrology has been shaped by a powerful, and often counter-intuitive, dynamic of large, opposing forces. While previous research has correctly identified the Kapchagay Reservoir's impoundment as the primary cause of the lake's decline in the 1970s and 80s (Duan et al., 2021; Kezer & Matsuyama, 2006), our detailed attribution provides a more nuanced understanding. We demonstrate that this human-induced water withdrawal was so immense (-9.21)




km<sup>3</sup>) that it completely masked a significant, climate-driven potential for a wetter regime (+6.13 km<sup>3</sup>), a phenomenon largely overlooked in prior assessments. This finding underscores that the era's ecological crisis was even more severe than the observed runoff decline alone would suggest, as it unfolded against a favorable climatic backdrop.

The most recent period (post-1991) reveals a critical evolution in the basin's vulnerability. Our analysis shows that while direct human impact has lessened, the basin's apparent stability is heavily subsidized by an exceptionally favorable climatic trend, characterized by increased rainfall and decreased potential evapotranspiration. This aligns with recent findings of increasing precipitation in parts of Central Asia (Jin et al., 2024), but our study is the first to quantify how this "climatic boon" (+10.80 km<sup>3</sup>) has been almost entirely neutralized by ongoing human water use (-11.36 km<sup>3</sup>). This exposes a fragile equilibrium: the current stability of Lake Balkhash is not a sign of intrinsic resilience or successful water management, but rather a coincidental and likely temporary stalemate between massive, opposing forces. This finding challenges the narrative of a simple "recovery" and instead highlights a state of heightened, masked vulnerability. Any future reversal of the recent favorable precipitation or PET trends could rapidly expose the system to the full, unbuffered impact of this lost storage and sustained human demand. To further probe the precarious equilibrium identified in Period 3 and to underscore the system's vulnerability, we extended our integrated modeling framework to project future changes in Lake Balkhash's water level through 2100 under three distinct Shared Socioeconomic Pathways (SSPs): a low-emissions sustainability scenario (SSP1-2.6), a medium-high emissions scenario (SSP3-7.0), and a very high-emissions, fossil-fueled development scenario (SSP5-8.5). Parameters were referenced from P3, and climate-driven data were obtained from NEX-GDDP-CMIP6. To simulate future land-use changes, we referenced the research of (Guo et al., 2015), simulating future land-use conditions based on the land expansion rate over the past 20 years and the maximum expansion condition of 15%.

Figure 11 Changes in lake water levels under three future scenarios






The results as depicted in Fig. 11, reveal a concerning trajectory across all considered futures, unequivocally demonstrating that the apparent stability of the late 20th and early 21st centuries is a fragile and transient condition. Under the most optimistic scenario (SSP1-2.6), Lake Balkhash is projected to experience a slow but steady decline, with its water level falling by approximately 2.5 meters from its 2020 level by 2100. This indicates that even with aggressive global climate mitigation, the combined pressures of existing human water demand and the legacy of cryospheric decline will likely overwhelm the system's resilience. In stark contrast, the medium to high-emissions scenarios (SSP3-7.0 and SSP5-8.5) portend a much more severe fate. Under these pathways, the lake experiences a pronounced and accelerated decline, particularly after 2040s. Historically, the lowest water level of Lake Balkhash was 340.52 meters, a period marked by severe ecological conditions. Under the medium emissions scenario, this water level would be reached by the 2070s of this century, while under the high emissions scenario, it would occur as early as the 2050s of this century. These scenarios starkly illustrate that the "climatic boon" of increased precipitation and reduced PET that subsidized the lake's water balance in recent decades is not guaranteed to continue. As this favorable climatic trend wanes or reverses under future warming, the persistent and powerful negative impacts of human water demand and the systematic loss of glacial storage—previously masked—are set to become the dominant drivers, precipitating a rapid contraction of the lake. These future projections confirm our central finding: the current state of Lake Balkhash is not one of recovery, but of a masked vulnerability, sustained by a temporary climatic subsidy that is highly likely to diminish, exposing the ecosystem to severe and potentially irreversible consequences.

While HAAF framework demonstrated robust performance and provided novel insights, certain limitations and uncertainties should be acknowledged. First, the Budyko framework, despite its strengths, simplifies complex landscape processes into a single parameter (n). The positive contribution attributed to "human activity" (the  $\triangle$ n term) in our component analysis likely captures complex, secondary land-surface feedbacks (e.g., irrigation return flows, land use changes) that are difficult to disentangle from the primary impact of direct water withdrawal. Future work could employ more process-detailed models or isotopic tracers to further partition these human impacts. Second, although we used the highest-quality, long-term datasets available, uncertainties persist in historical meteorological forcings and reconstructed naturalized flows, particularly in the early 20th century. Our integrated validation of the lake's water balance provides strong confidence in our overall results, but continued efforts to improve historical data rescue and reconstruction are vital.

#### 5. Conclusion

This study developed and implemented a novel attribution framework HAAF, integrating PIML models with the Budyko framework and a lake water balance model, to quantitatively separate the impacts of climate change and human activities on the centennial hydrology of the Lake Balkhash basin. Our primary objective was to provide a comprehensive, quantitative explanation for the lake's historical water level dynamics by tracing the drivers of change from the catchment to the lake terminus. Our key findings are as follows:

https://doi.org/10.5194/egusphere-2025-4778 Preprint. Discussion started: 19 November 2025

© Author(s) 2025. CC BY 4.0 License.



(1) The HAAF successfully reconstructed the naturalized and human-impacted streamflow with high fidelity (KGE > 0.75, 450 |PBIAS| < 10% for most stations), establishing a robust foundation for attribution analysis. The final integrated validation confirmed that our framework accurately closes the lake's water balance, with simulated volume changes strongly correlating with satellite-derived observations (R = 0.86).</p>

(2) At the catchment scale, our attribution of runoff changes revealed a powerful dynamic of opposing forces. During the intensive intervention period (1970-1990), direct human activities were the overwhelming driver of runoff decline, causing a reduction of -9.21 km<sup>3</sup>. This immense anthropogenic pressure completely masked a significant climate-driven potential for a wetter regime (+6.13 km<sup>3</sup>). Subsequently, during the recent period (1991-2024), the basin's hydrology entered a state of fragile equilibrium, where a massive climate-driven potential for runoff increase (+10.80 km<sup>3</sup>) was almost entirely counteracted by the sustained impact of human water use (-11.36 km<sup>3</sup>).

(3) These catchment-scale dynamics directly governed the lake's response. The apparent stability of Lake Balkhash in recent decades was not a sign of resilience but was heavily subsidized by a highly favorable climatic trend that buffered the system against persistent anthropogenic water stress. However, our future projections under multiple SSP scenarios reveal that this climatic buffer is likely transient. As the favorable climatic conditions wane, the underlying pressures from human activities and cryospheric decline are projected to dominate, leading to a rapid and severe drop in the lake's water level. This demonstrates that the current equilibrium is precarious and masks a profound long-term vulnerability.

In conclusion, our research reveals that the apparent recent stability of Lake Balkhash is not a sign of systemic recovery but a fragile illusion sustained by a transient climatic boon—a state of "masked vulnerability." This study underscores the critical need for effective water management strategies that look beyond recent trends and account for the underlying, competing drivers of the system. The HAAF framework presented here provides a powerful tool for diagnosing similar complex hydroclimatic systems globally, offering a crucial scientific basis for sustainable water resource management in an era of accelerating environmental change.

*Data availability.* All underlying data used in this research study are openly available. The sources are mentioned in Sect. 2.2. Model outputs as well as code may be made available by request to the corresponding author.

Author contributions. JW designed the study. RY performed the analysis and wrote the manuscript. All authors commented on the manuscript.

Competing interests. The contact author has declared that none of the authors has any competing interests.

*Financial support:* We thank the CAS Research Center for Ecology and Environment of Central Asia for assistance with this work. This study was supported by the National Natural Science Foundation of China (U2003202), and the Third Xinjiang Scientific Expedition and Research Program (Grant No. 2022xjkk070201).

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
