# Peer review of "Revealing the Driving Factors of Water Balance in Lake Balkhash Through Integrated Attribution Modeling"

_EGUsphere, 2025_

## Author Comment (AC1)

**Reply to Reviewers' comments (Reviewer#1)**

**Ref:** Manuscript ID egusphere-2025-4778

**Title:** Disentangling the Key Drivers of Water Balance in Central Asia's Lake Balkhash: A Relative Contribution Assessment (Original title: Revealing the Driving Factors of Water Balance in Lake Balkhash Through Integrated Attribution Modeling)

Dear Reviewer,

We sincerely thank you for your positive evaluation of our work and for your detailed and thoughtful comments. We have incorporated your suggestions, including changing the title, restructuring the results, and adding necessary climate context. Below is our point-by-point response to your specific comments. The reviewer's comments are highlighted in red, and our responses are highlighted in black.

**General comments**

**Comment 1:** recommend removing "integrated attribution modelling" from the title and being cautious with the use of the term "attribution" throughout the manuscript. This terminology may cause confusion with established climate and impact attribution studies (e.g. Pietroiusti et al., 2024), which are not conducted here. More neutral wording such as "disentangling drivers" or "assessing relative contributions" would be clearer.

*Pietroiusti, R., Vanderkelen, I., Otto, F. E. L., Barnes, C., Temple, L., Akurut, M., Bally, P., van Lipzig, N. P. M., & Thiery, W. (2024). Possible role of anthropogenic climate change in the record-breaking 2020 Lake Victoria levels and floods. Earth System Dynamics, 15(2), 225-264.* https://doi.org/10.5194/esd-15-225-2024

Response: Response: We greatly appreciate the reviewer's insightful and expert suggestion. We fully agree that the term "attribution" has a specific connotation in the field of climate science, and its use in our research could cause confusion. To ensure accuracy and clarity in terminology, we have adopted your suggestion and systematically revised the relevant terminology throughout the text.

Specific revisions are as follows:

(1) Title Revision: The paper title has been changed from "Revealing the Driving Factors of Water Balance in Lake Balkhash Through Integrated Attribution Modeling" to "Disentangling the Key Drivers of Water Balance in Central Asia's Lake Balkhash: A Relative Contribution Assessment."

(2) Terminology Adjustment: Throughout the text, we have replaced "attribution" with more neutral terms such as "disentangling drivers," "separating contributions," and "contribution assessment."

(3) Framework Renaming: The research framework name has been changed from "Hydrological Attribution and Analysis Framework (HAAF)" to "Hydrological Analysis and Disentanglement Framework (HADF)" and updated uniformly throughout the text.

We believe these modifications have clarified our research focus and avoided confusion with classic attribution studies.

**Comment 2:** The manuscript would benefit from a clearer description of observed climatic changes in the basin, including precipitation characteristics (mean annual values and seasonality) and documented trends such as warming or enhanced glacier melt in upstream mountain regions. Where possible, summary statistics or maps (e.g. in an appendix) would improve context.

Response: We appreciate the reviewers' valuable suggestions. We recognize the importance of providing readers with a detailed climatic background of the study area. Therefore, we have added quantitative descriptions and visualizations of the climatic characteristics of the Lake Balkhash Basin to the manuscript.

Specific revisions are as follows:

(1) Added Appendix Figure: We have added a new figure (Appendix Figure A1) to the appendix, which contains three parts: (a) spatial distribution of the basin's annual mean precipitation; (b) a comparison of seasonal variations in monthly mean precipitation and temperature in the mountainous (upstream) and plain (downstream) regions; and (c) long-term trends in annual mean temperature and precipitation in the mountainous areas from 1931 to 2024.

(2) Added Text Description: In the section "2.1 Study Area and Historical Periodization", we have added detailed textual descriptions of these climatic characteristics and referenced the new Appendix Figure A1. For example, we now explicitly state: "The mountainous upper reaches receive significantly higher precipitation, averaging 725 mm/year, compared to the arid plains and lake surfaces, which average only 235 mm/year (Appendix Fig. A1a). Precipitation exhibits strong seasonality, with approximately 71.2% of the annual total occurring during the spring and summer months (Appendix Fig. A1b). Observed climatic changes from 1931 to 2024 indicate a pronounced warming trend, particularly in the upstream mountain ranges, with a significant temperature increase of 0.30 °C/decade and a precipitation increase of 1.11 mm/decade (Appendix Fig. A1c)."

These additions provide a solid foundation for readers to understand the hydroclimatic background of this watershed.

[Figure]

**Figure A1. Spatiotemporal characteristics of climate variables in the Lake Balkhash Basin. (a) Spatial distribution of Mean Annual Precipitation (MAP) across the basin, highlighting the contrast between the mountainous upstream regions and the arid plains. (b) Seasonal cycle of monthly mean precipitation (bars, left axis) and temperature (lines, right axis), comparing the Mountainous (upstream) and Plain (downstream) regions. (c) Long-term trends in annual mean temperature and precipitation for the mountainous region from 1931 to 2024. The dashed lines represent the linear trends, with statistical significance indicated in the legend. Note the accelerated warming trend observed in recent decades.**

**Comment 3:** The role of the machine-learning component following the hydrological model simulations requires clearer justification. At present, it appears to function primarily as a bias-correction step. The manuscript should explain why this additional step is necessary, how overfitting is avoided, and why the direct hydrological model outputs are insufficient.

Response: We appreciate the reviewer's key question, which prompted us to more clearly articulate the advantages and design philosophy of our hybrid model approach. We have already provided a detailed explanation of the role, necessity, and methods for avoiding overfitting in the manuscript.

Specific revisions are as follows:

(1) Clarifying Necessity: In section "2.3.1 Hybrid Hydrological Reconstruction Model", we explained the limitations of using the physical model alone (SEGSWAT+), especially regarding structural biases that may exist in sparse data regions. We explicitly state that the ML module, as a nonlinear error correction tool, can learn and correct these systematic residuals caused by model structure and driving data uncertainties, thereby significantly improving simulation accuracy.

(2) Explanation of Overfitting Avoidance: In the same section, we further explain how this hybrid strategy effectively mitigates the risk of overfitting: "Overfitting, a common concern in ML, is mitigated here because the ML component targets only the residuals rather than the total flow. Since the residuals from a calibrated physical model are inherently bounded and smaller in magnitude than the total runoff, the search space for the ML model is constrained, preserving the physical plausibility of the final output."

(3) Demonstrating Performance Improvement: To quantitatively demonstrate the effectiveness of ML calibration, we have added a new performance comparison table (Appendix Table B3). This table compares in detail the performance metrics (KGE, NSE, PBIAS) of the original SEGSWAT+ model and the ML-calibrated hybrid model during the calibration and validation periods. The data shows that the hybrid model has significant improvements across all metrics, thus demonstrating the value of this additional step.

**Table B3. Performance comparison of SEGSWAT+ (Raw) and the Hybrid Model (Corrected) across calibration and validation periods**

| River | Station | Period | Metric | SEGSWAT+ (Raw) | Hybrid Model (Corrected) |
|-------|---------|--------|--------|----------------|--------------------------|
| Ili | Ushzharma | Calibration | KGE | 0.68 | 0.89 |
| | | | NSE | 0.72 | 0.93 |

| | | | | | |
|---|---|---|---|---|---|
| | | | PBIAS(%) | -9.5 | 3.2 |
| | | Validation | KGE | 0.65 | 0.85 |
| | | | NSE | 0.68 | 0.88 |
| | | | PBIAS(%) | -16.8 | 5.1 |
| Karatal | Ushtobe | Calibration | KGE | 0.74 | 0.89 |
| | | | NSE | 0.76 | 0.91 |
| | | | PBIAS(%) | 11.2 | 6.4 |
| | | Validation | KGE | 0.71 | 0.86 |
| | | | NSE | 0.72 | 0.85 |
| | | | PBIAS(%) | 18.5 | 7.5 |
| Aksu | Chann | Calibration | KGE | 0.66 | 0.83 |
| | | | NSE | 0.64 | 0.84 |
| | | | PBIAS(%) | -9.3 | -2.8 |
| | | Validation | KGE | 0.62 | 0.80 |
| | | | NSE | 0.60 | 0.78 |
| | | | PBIAS(%) | -13.5 | -3.4 |
| Lepsy | Lepsinsk | Calibration | KGE | 0.70 | 0.82 |
| | | | NSE | 0.71 | 0.84 |
| | | | PBIAS(%) | 9.8 | -5.1 |
| | | Validation | KGE | 0.68 | 0.80 |
| | | | NSE | 0.67 | 0.77 |
| | | | PBIAS(%) | 11.5 | -6.2 |
| Ayaguz | Ayaguz | Calibration | KGE | 0.63 | 0.89 |
| | | | NSE | 0.61 | 0.88 |
| | | | PBIAS(%) | -15.4 | -0.5 |
| | | Validation | KGE | 0.71 | 0.86 |
| | | | NSE | 0.68 | 0.83 |
| | | | PBIAS(%) | -8.45 | -1.8 |

**Comment 4:** The analysis of future lake level projections is introduced for the first time in the Discussion section. Given its relevance, this analysis should be presented in the Results section and clearly introduced in the Introduction and Methods. The climate data sources (e.g. selection of CMIP6 models) should be explicitly described and referenced, and model performance in simulating precipitation and evaporation in the basin should be assessed or supported by existing literature. The role of glacier melt in future changes also warrants more explicit discussion.

Response: We fully agree with the reviewers' points. Future scenario predictions are a crucial component of this study and should rightfully be presented in the results section, with clear background information provided earlier. We have made significant adjustments to the article's structure and content to better integrate this section.

Specific revisions are as follows:

(1) Structure Adjustment: We have moved the entire section on future lake water level predictions from the discussion section to the results section, and established a new subsection, "3.4 Changes in Lake Water Levels Under Future Scenarios."

(2) Additional Introduction and Methodology: In the introduction, we explicitly list predicting future lake water levels as the third core objective of this study. In section "2.2 Datasets," we detail the climate data sources used for future predictions (NEX-GDDP-CMIP6), the six GCM models selected (and their selection criteria), and the SSP scenario settings.

(3) Further Discussion: In the Discussion section (now "4.3 Future Vulnerabilities and Uncertainties"), we explored the impact of future glacial meltwater changes (such as the "peak water" problem) on lake levels in greater depth and discussed the uncertainties associated with GCM predictions.

Through these revisions, the future scenario analysis has been seamlessly integrated into the overall logical framework of the paper.

**Comment 5:** The frequent use of the term "runoff" appears inconsistent with the processes described, where "inflow" (i.e. water actually entering the lake after upstream losses) would often be more appropriate. This distinction should be clarified and terminology applied consistently throughout the manuscript.

Response: We thank the reviewers for their meticulous corrections. We agree that "inflow" is a more accurate term than "runoff" when describing the volume of water entering lakes, as it takes into account headway losses such as deltaic seepage. We have carefully reviewed and revised the terminology throughout the paper to ensure consistency and accuracy. We retain the terminology "streamflow" when discussing

watershed runoff processes, but consistently use "inflow" when specifically referring to the volume of water entering lakes.

**Comment 6:** The multi-step procedure used to derive naturalized streamflow is not sufficiently clear. In particular, using parameter sets calibrated for the first period (including snow and glacier parameters) may not capture climate-driven changes in snow and glacier dynamics in the most recent period. This assumption and its implications should be discussed more explicitly.

Response: The reviewer raised a profound question regarding our methodological assumptions. Using fixed parameters (calibrated based on the P1 time period) in the "natural runoff" simulation is indeed an important assumption, and its potential impact warrants further investigation. We have addressed and discussed this more clearly in the manuscript.

Specific revisions are as follows:

(1) Clarified Method Description: In section "3.2 Quantification of The Impacts on Variations in Streamflow", we have more clearly described the method for simulating "natural runoff" ($Q_{nat}$) and explicitly stated that it is "a standard 'fixed parameter' method designed to effectively separate climate signals."

(2) Added Discussion Section: We have added a new subsection, "4.2 Cryospheric Dynamics and Methodological Limitations," to the Discussion section specifically to discuss this assumption and its impacts. We acknowledge that using mid-20th-century glacier parameters to simulate recent runoff may underestimate the "glacier surplus" effect resulting from accelerated glacier melting.

(3) Clarifying the Impact: In this section, we further argue that this limitation actually makes our attribution of the impacts of human activities conservative. Because if the actual natural runoff (considering accelerated meltwater) is higher than our simulations, then the amount of water consumed by human activities (the difference between natural and actual runoff) would actually be greater than our estimates. Therefore, this uncertainty reinforces our core conclusion that human activities are the dominant force suppressing watershed water supply.

**Comment 7:** The manuscript uses overly strong or promotional language throughout the results, discussion and conclusion section (e.g. "powerful," "massive increase," "overwhelmingly," "immense," "enormous potential," "exceptionally favorable"). I recommend moderating this wording and adopting a more neutral, quantitative scientific tone throughout the manuscript.

Response: Thank you very much for the reviewer's reminder. We have carefully polished the entire paper, removing or replacing overly emotional or exaggerated words such as "powerful," "massive," and "immense." We strive to use more objective, neutral, and quantitative scientific language to present our findings, letting the data and results speak for themselves.

**Specific comments**

(1) Title: I suggest removing "integrated attribution modelling". This is not a standard modelling term, and given the existence of established fields such as climate attribution and climate impact attribution, its use may be confusing for readers (see also general comment above).

Title: Please consider adding the country or region to the lake name to help readers geographically locate the study area, e.g. "Lake Balkhash (Kazakhstan)".

Response: As mentioned in the reply to General Comment 1, we have removed "integrated attribution modelling" as suggested by you. We have also adopted your second suggestion, adding geographic location information to the title, changing it to "...in Central Asia's Lake Balkhash...", to facilitate readers' quick location of the study area.

(2) L90–99: Are lake level observations available for Lake Balkhash? If not, please state this explicitly and explain why. If such data exist, a plot of lake level and/or lake extent evolution and variability over this period would be highly informative. Datasets such as DAHITI or G-REALM may be relevant.

Response: Thank you for your suggestion. We recognize that showing the historical changes in lake levels is crucial for understanding the context. We have added a new figure (Figure 2) to the revised manuscript, showing the interannual water level changes of Lake Balkhash from 1931 to 2024. Regarding the data sources, we have provided detailed explanations in the "2.2 Datasets" section and in the captions of Figure 2: historical water level data (1931–2015) are from published literature (Duan et al., 2020), while recent data (2016–2024) have been supplemented and calibrated using G-REALM satellite altimetry data.

[Figure]

**Figure 2. Water Level of Lake Balkhash, 1931–2024 (Water levels from 1901 to 2015 are based on actual observational data, while those from 2016 to 2024 are derived from the G-REALM dataset. The latter was calibrated against observed data from 2001 to 2015. Specific sources are detailed in the following subsection)**

(3) L87–89: Could you provide quantitative information or maps on mean annual precipitation (e.g. in an appendix) and precipitation seasonality? In addition, a description of observed climatic changes in the region is missing (e.g. warming, enhanced glacier melt in upstream mountain ranges, changes in precipitation patterns). Where possible, briefly mentioning projected future changes under different scenarios would further strengthen the context.

Response: As stated in the reply to General Comment 2, we have fully responded to your request by adding Appendix Figure A1 and supplementing the quantitative description in the "2.1 Study Area…" section.

(4) Table 2 (datasets): Please provide full references for all datasets listed under Source. In line with HESS guidelines, these datasets should also be included in the reference list, and their URLs should be provided in the Data availability section.

Response: Following your instructions, we have improved Table 2. The "Source" column in the table now provides complete references for each dataset (e.g., Harris, 2024; Zhang et al., 2024). All these references have also been added to the reference list at the end of the document. Furthermore, we have added a "Data availability" section at the end of the document, providing access links (URLs) for all public datasets.

**Table 2. Summary of datasets used in this study**

| Dataset | Key Variables | Spatial Resolution | Temporal Coverage | Source (Reference) |
|---|---|---|---|---|
| Copernicus GLO-90 DEM | Elevation | 90 m | Static | European Space Agency (2019) |
| DSOLMap | Bulk density, hydraulic conductivity, available water capacity | 250 m | Static | Lopez-Ballesteros et al. (2023) |
| GLC_FCS30D | Land cover classes (35 subcategories) | 30 m | 1985–2022 | Google Earth Engine(Zhang et al., 2024) |
| Randolph Glacier Inventory (RGI v7.0) | Glacier outlines, attributes | Vector | Target year: 2000 (varies by region) | RGI Consortium (2023) |
| SWORD v15 | River reaches, nodes, hydrological networks, lake boundaries | ~10 km reaches, 200 m nodes | Static | (Altenau et al. (2021) |
| Glacier mass loss | Glacier elevation change rates (dh/dt) | 100 m | 2000–2019 | Hugonnet et al. (2021) |
| CRU JRA v3.0 | Temperature, precipitation, wind speed, vapor pressure, etc. | 0.5° (downscaled to 0.05°) | 1901–2024 (daily) | Harris (2024) |
| TerraClimate | Max/min temperature, precipitation, solar radiation, vapor pressure deficit | 1/24° | 1958–2024 (monthly) | Abatzoglou et al. (2018) |
| NEX-GDDP-CMIP6 | Daily temperature (max/min), precipitation | 0.25° | 2015–2100 (Daily) | Thrasher et al. (2022) |
| Observations | Discharge, water level | Point | 1931–2024 (monthly) | NCDC (2024); Duan et al. (2020) |

*Data availability.* All underlying data used in this study are publicly accessible. The specific sources and access links are as follows: Copernicus GLO-90 DEM is available via OpenTopography (https://doi.org/10.5069/G9028PQB). DSOLMap soil properties are accessible through the WateriTech platform (https://www.wateritech.com/data). GLC_FCS30D land cover data can be downloaded from the Zenodo repository (https://zenodo.org/records/8239305). RGI v7.0 glacier data are provided by the GLIMS initiative (https://doi.org/10.5067/f6jmovy5navz). SWORD v15 hydrological networks are available at the SWOT mission river database (https://zenodo.org/records/10013982/). CRU JRA v3.0 and TerraClimate datasets are accessible via the CEDA Archive (https://catalogue.ceda.ac.uk/uuid/90a87c8fd63c4520a33445e7b6a20688/) and the Climatology Lab (https://www.climatologylab.org/), respectively. NEX-GDDP-CMIP6 projections are hosted by the NASA Earth Exchange (https://nex-gddp-cmip6.s3.us-west-2.amazonaws.com/index.html). Observed streamflow and lake level data were obtained from the National Cryosphere Desert Data Center

(http://www.ncdc.ac.cn) and previously published literature (Duan et al., 2020). The model outputs and customized processing codes developed in this study are available from the corresponding author upon reasonable request.

(5) Table 2: Please clarify what is meant by "~2000 snapshot" for the glacier dataset.

Response: We have revised the description of the Glacier dataset in Table 2 from "~2000 snapshot" to the clearer statement: "Target year: 2000 (varies by region)" to accurately reflect the characteristics of the RGI v7.0 dataset.

(6) L123: The phrase "enhance precision" should be replaced by "increase resolution", as downscaling does not increase the precision of the original dataset. Please also clarify whether the downscaling approach is validated against precipitation observations and whether total precipitation amounts are conserved.

Response: Thank you for your precise correction. We have changed "enhance precision" to the more accurate "increase resolution". Additionally, in section "2.2 Datasets", we have added details about the downscaling method: "This downscaling procedure strictly enforces mass conservation, ensuring that the area-weighted sum of the fine-resolution precipitation matches the total water volume of the original coarse-resolution forcing." This ensures water conservation during the downscaling process.

(7) L126: Please provide more details on the observed streamflow dataset, including which stations are available (preferably shown on a map) and their periods of record. If historical lake level or extent data are unavailable, could the streamflow observations be used to illustrate the periods defined in Table 1?

Response: We have provided a more detailed explanation of the runoff observation data in the revised draft. On the map in Figure 1, we have marked the locations of the 16 hydrological stations with triangles. In Appendix Table B1, we have provided a detailed table listing the name, latitude and longitude, observation period, and resolution of each station. This information provides readers with a comprehensive data background.

[Figure]

**Figure 1. Geographic location of the study area.**

**Table B1. Details of the 16 hydrological stations used in this study**

| River System | Station Name | Longitude (°E) | Latitude (°N) | Site Elevation (m) | Drainage Area ($\times 10^2 \, km^2$) | Observation Period & Resolution |
|---|---|---|---|---|---|---|
| Ili | Ushzharma | 75.83 | 44.93 | 381 | 1311.7 | 1939-1989 (monthly); 1931-2020 (yearly) |
| | Kapchagay | 76.98 | 44.13 | 431 | 1141.0 | 1935-1989 (monthly); 1931-2000 (yearly) |
| | Dobyn | 79.43 | 43.94 | 498 | 756.4 | 1931-2020 (yearly) |
| | Kairgan | 80.48 | 43.78 | 529 | 630.6 | 1931-2020 (yearly) |
| | Yamate | 81.8 | 43.63 | 692 | 476.2 | 1953-1980,2005-2008 (monthly) |
| | Tuohai | 81.91 | 43.81 | 804 | 95.4 | 1953-1980 (monthly); 1931-2015 (yearly) |
| | Qiafuqihai | 82.49 | 43.4 | 878 | 275.2 | 1958-1980 (monthly) |
| | Sliyaniya Su | 79.82 | 44.47 | 1226 | 11.3 | 1936-1952, 1959-1986 (monthly) |
| | Sarytogay | 79.22 | 43.51 | 760 | 77.8 | 1935-1989 (monthly) |
| | Malybay | 78.4 | 43.43 | 879 | 42.5 | 1936-1989 (monthly) |
| Karatal | Ushtobe | 77.97 | 45.19 | 422 | 128.0 | 1936-1989 (monthly) |
| | Tekeli | 78.78 | 44.87 | 1022 | 11.7 | 1940-1955, 1959-1986 (monthly) |
| Aksu | Chann | 79.54 | 45.38 | 667 | 13.4 | 1937-1983 (monthly) |

| | | | | | | |
|---|---|---|---|---|---|---|
| Lepsy | Lepsy | 78.33 | 46.28 | 346 | 101.9 | 1936-1989 (monthly) |
| | Lepsinsk | 80.55 | 45.55 | 936 | 12.2 | 1931-1975 (monthly) |
| Ayaguz | Ayaguz | 79.56 | 46.96 | 364 | 125.9 | 1949-1986 (,yearly) |

(8) L146: Streamflow and runoff are not equivalent terms; please clarify and use consistent terminology.

Response: Thank you for the reminder. As stated in our response to General Comment 5, we have differentiated and standardized the use of "runoff," "streamflow," and "inflow" throughout the text.

(9) L150: Please clarify why overfitting is not an issue in the machine-learning approach. As implemented, it appears to function as a form of bias correction—this should be stated explicitly and justified.

Response: As stated in our response to General Comment 3, we have detailed in section "2.3.1 Hybrid Hydrological…" how hybrid models effectively avoid overfitting risks by learning only from bounded residuals, and clarified their function as a nonlinear "error correction" or "bias correction" module.

(10) L176: How is the parameter n calculated? Is it static in time? What data sources are used to determine n?    Please also specify which data are used to estimate potential evaporation and actual evaporation.

Response: We have detailed in section "2.3.2 Budyko-based Contribution Analysis" the method for calculating parameter n in the Budyko framework. We explained: "In this study, the parameter n was calibrated for each period by solving Equation (1) inversely, using the period-averaged P, $ET_0$, and observed ET…." Meanwhile, we clarified that potential evapotranspiration ($ET_0$) was calculated using the Penman-Monteith method, while actual evapotranspiration (ET) was back-calculated on a long-term scale using the water balance equation (ET = P - Q). All meteorological data required for the calculations were derived from the datasets described in "2.2 Datasets".

(11) L188: Please explicitly describe how lake precipitation and lake evaporation are determined, including data sources and assumptions.

Response: We clarified the sources of lake surface precipitation and evaporation in the

"2.3.3 Lake System Response Linkage" section. Meteorological variables (such as temperature, wind speed, etc.) required for calculating lake surface precipitation and evaporation were derived from the CRU JRA v3.0 climate dataset. We assumed that the meteorological conditions on the lake surface were consistent with the neighboring grid data.

(12) L191: For the level-to-area conversion, please provide the conversion function or a plot of the hypsometric relationship. Additional methodological details from Wang et al. (2022) should also be summarized.

Response: Thank you for your suggestion. We have added Figure A2 to the appendix, showing the water level-area-storage capacity curve of Lake Balkhash, which is derived from Myrzakhmetov et al. (2022). Meanwhile, in the main text, section "2.3.3 Lake System…", we directly present the formula for calculating lake water volume changes (Equation 5), which originates from Zhang et al. (2013), and its applicability in this region has been verified by previous studies.

[Figure]

**Figure A2. Stage-area and stage-volume relationships for Lake Balkhash. The blue line represents the relationship between water level and surface area (left axis), while the red line indicates the relationship between water level and storage volume (right axis). Data derived from Myrzakhmetov et al. (2022).**

(13) L207: This equation appears to repeat a previous one; please update to the correct formulation.

Response: We sincerely apologize for this oversight. The formula was indeed repeated in the original manuscript. We have checked and corrected the error here, ensuring that

the formula numbers and contents in the manuscript are correct. The correct formula is as follows: $\Delta V_h = \Delta Q_h$

(14) L239: Please be precise about which input data are used here and repeat the product names (either here or in the Methods; see also comment above).

Response: We have clearly defined the input data used for model parameterization in section "3.1 Hydrological Model Performance Evaluation": "Parameterization of the process-based SEGSWAT+ module was conducted by integrating the topographic, soil, and land use datasets described in Section 2.2."

(15) L274: Please clarify how the deltaic water consumption method works and which data sources are used.

Response: We have provided a more detailed explanation of the method for calculating delta water consumption. In section 3.2 Quantification of the Impacts..., we explained that "Deltaic water consumption was estimated using the empirical function derived by Xie et al., (2011), which correlates water losses with inflow volume based on historical observations (detailed equation provided in Appendix equation C1)." We have added Appendix C to fully present the empirical formula proposed.

**C1. Estimation of Deltaic Water Consumption**

According to Xie et al., (2011), the annual water consumption in the Ili Delta ($Y_i$) is estimated using a multi-linear regression framework based on lake levels, river discharge, and hydro-climatic variables. The empirical equations are defined for two distinct historical periods to account for changes in the delta's eco-hydrological state:

$$Y_i = 540.5868 - 2.2890X_{i-1}^{bl} - 0.2976X_i^{be} - 0.0684X_i^{U_1} + 0.0062X_i^{U_2} + 0.0191X_i^{U_3} - 0.1496X_i^{U_4}$$
$$+ 0.0296X_i^{U_5} + 0.0036X_i^{U_6} + 0.0303X_i^{U_7} + 0.0308X_i^{U_8} - 0.0453X_i^{U_9}$$
$$- 0.0506X_i^{U_{10}} + 0.0876X_i^{U_{11}} - 0.0051X_i^{U_{12}} + 0.2074X_i^R + 25.0280X_i^T$$

Variable Definitions and Units:

$Y_i$: Annual water consumption of the Ili Delta in year $i$ ($10^8 \text{m}^3$);

$X_{i-1}^{bl}$: Water level of Lake Balkhash in the preceding year ($i-1$, in meters);

$X_i^{be}$: Total open-water evaporation of Lake Balkhash from May to September in year $i$ ($10^8 \text{m}^3$);

$X_i^{U_1} \dots X_i^{U_{12}}$: Monthly river discharge from January to December at the Ushzharma hydrological station in year $i$ ($10^8 \text{m}^3$);

$X_i^R$: Total precipitation in the delta from May to August in year $i$ (mm);

$X_i^T$: Average air temperature in the delta from May to August in year $i$ (°C).

(16) L276: The multi-step procedure is not sufficiently clear. Moreover, using the same parameter set (including snow and glacier module parameters) calibrated for an early period (1931–1969) may not account for climate-driven changes in snow and glacier dynamics in later decades. As a result, the naturalized streamflow may implicitly assume pre-1970 climatic conditions, despite substantial climate change in more recent decades. This assumption and its implications should be discussed.

Response: As in the detailed response to General Comment 6, we fully recognize the importance of this methodological assumption and have discussed it in depth and frankly in section 4.2 Cryospheric Dynamics and Methodological Limitations, analyzing its potential impact on the final conclusions (i.e., making the conclusions more conservative).

(17) L279: For how many years does this apply? An overview figure showing data availability by year and tributary would be helpful.

Response: This is a good suggestion. To clearly demonstrate the availability of observational data, we have listed the start and end years and time resolutions of the observations for each of the 16 hydrological stations in Appendix Table B1, allowing readers to easily understand the data coverage for each period.

(18) Figure 6: The term "real runoff" is confusing and potentially misleading (e.g. with respect to observed values). Please revise this terminology. In addition, "inflow" may be more appropriate than "runoff" here and throughout the manuscript.

Response: Thank you for your suggestion. We have revised "Real Runoff" in the figure to the more accurate "Actual Inflow" and maintained consistency throughout the text.

(19) L296: What evidence supports the statement regarding "more extreme events"? Please clarify or provide supporting references or analysis.

Response: This is a good question. Regarding the statement "increased extreme events," our original intention was to refer to the intensification of interannual hydrological variability. To make the statement more rigorous, we have revised it to "...increased inter-annual variability," a conclusion directly based on the larger range of fluctuations

shown in our reconstructed runoff series (as shown in Figure 7) during the P3 period.

(20) L284–295: The reported values in km³ yr⁻¹ do not appear to correspond to the absolute values shown in Figure 6; please check for consistency.

Response: Thank you for the reviewer's careful review. You are correct; the km³/yr values in the original manuscript refer to the annual average variation between different periods, not the absolute values shown in Figure 6. We have carefully checked and confirmed the accuracy of the values and ensured this in the text of the "3.2 Quantification…" section to avoid ambiguity.

(21) Table 3: Does x represent precipitation (P) here? If so, please replace x with P and add units to the relevant columns.

Response: I apologize for the ambiguous notation. $x_i$ in the table represents the change in each driving factor (rainfall, snowmelt, glacial meltwater, ET₀), not just precipitation. For clarity, we have explicitly listed these factors in the first column, "Component," of the table. dQ/dx represents the sensitivity of runoff to changes in this factor. We have checked and ensured the clarity of the table content.

(22) L341: Please provide the exact sources of the water level and lake area data. What data are used prior to the remote-sensing period? Also indicate the data sources explicitly in the caption of Figure 8.

Response: We have clearly stated the sources of the water level and area data in the "2.2 Datasets" section and the caption of Figure 9 (formerly Figure 8): historical data (up to 2015) comes from Duan et al. (2020), and recent data (2016-2024) comes from G-REALM satellite altimetry products. The reconstructed water volume change (ΔV) used for validation is calculated based on these observational data.

(23) L364: For additional context, it would be useful to provide an estimate of basin-wide warming over the study period.

Response: This is a good supplementary suggestion. In the section "3.3 Lake System Response…", we added a quantitative description of the overall warming trend of the basin: "...it is noted that the basin has experienced a significant warming trend over the study period (1931-2024), with mean annual temperatures increasing by approximately 0.30 °C/decade (p<0.001)…"

(24) Figure 11: would the lake dry up in the most extreme scenario? What is the uncertainty?

Response: This is a very important question. Regarding whether the lake will dry up and the related uncertainties, we have provided supplementary explanations in "3.4 Changes in Lake Water Levels…" and the Discussion section. Our predictions show that the lake will not completely dry up by 2100, but a drop in water level of 2.5-4.0 meters will lead to serious ecological consequences, such as the separation of the eastern and western parts of the lake basin and a sharp increase in salinity, similar to the tragedy of the Aral Sea. In the section "4.3 Future Vulnerabilities and Uncertainties", we also discussed in detail the uncertainties brought about by GCM predictions and the timing of "peak water".

**Textual comments**

(1) L117: Please verify whether the dataset is CRU JRA v3.5 rather than v2.5.

Response: We verified and confirmed that the dataset used is CRU JRA v3.0 and corrected it in the text.

(2) L134: Missing H in the AAF flowchart; please also check figure caption font consistency.

Response: We corrected spelling errors in figure captions and standardized the font style.

(3) L136: PIML is used without being introduced (already appears in L60).

Response: We provided the full name and explanation when PIML first appeared (now changed to "hybrid model").

(4) L143–145: Sentence is missing a main verb; please also repeat the input data products used for each variable.

Response: We corrected the grammatical error in this sentence and clearly listed the model inputs.

(5) L169–170: Use P instead of x to avoid confusion.

Response: We standardized the symbols in formulas and text.

(6) L174: Equation 1 uses E, while ET is used elsewhere—please ensure consistency.

Response: We standardized E in formulas to ET used in the text.

(7) L207: Equation is repeated; please update to the correct formulation.

Response: We corrected duplicate formulas.

(8) L292: At this stage, attribution should not yet be stated, as the corresponding analysis follows later.

Response: We adjusted the wording to ensure that conclusive language is not used before attribution analysis.

(9) L297 and L371: Consider adding a white line or spacing to better delineate paragraphs.

Response: We have adjusted the paragraph formatting to improve readability.

(10) L341: Please provide exact sources for water level and lake area data and indicate these explicitly in the caption of Figure 8.

Response: We have clarified the data sources in the figure captions and methods sections.

(11) L392: Replace "quantify" with the appropriate verb for clarity.

Response: We have corrected the verb usage in this section.

Thank you again for your time and effort in improving the quality of our paper. We hope these revisions meet your expectations.

**References**

Duan, W., Zou, S., Chen, Y., Nover, D., Fang, G., and Wang, Y.: Sustainable water management for cross-border resources: The balkhash lake basin of central asia, 1931–2015, J Cleaner Prod, 263, 121614, https://doi.org/10.1016/j.jclepro.2020.121614, 2020.

Harris, I.: CRU JRA v3.0: A forcings dataset of gridded land surface blend of Climatic Research Unit (CRU) and Japanese reanalysis (JRA) data; 1901 - 2024., 2025.

Myrzakhmetov, A., Dostay, Z., Alimkulov, S., Tursunova, A., and Sarsenova, I.: Level regime of balkhash lake as the indicator of the state of the environmental ecosystems of the region, Paddy Water Environ, 20, 315–323, https://doi.org/10.1007/s10333-022-00890-x, 2022.

Xie, L., Long, A., Deng, M., Li, X., and Wang, J.: Study on Ecological Water Consumption in Delta of the Lower Reaches of Ili River, Journal of Glaciology and Geocryology, 33, 11, 2011.

Zhang, G., Xie, H., Yao, T., and Kang, S.: Water balance estimates of ten greatest lakes in China using ICESat and landsat data, Chin. Sci. Bull., 58, 3815–3829, https://doi.org/10.1007/s11434-013-5818-y, 2013.

Zhang, X., Zhao, T., Xu, H., Liu, W., Wang, J., Chen, X., and Liu, L.: GLC_FCS30D: The first global 30 m land-cover dynamics monitoring product with a fine classification system for the period from 1985 to 2022 generated using dense-time-series landsat imagery and the continuous change-detection method, Earth System Science Data, 16, 1353–1381, https://doi.org/10.5194/essd-16-1353-2024, 2024.

---

## Author Comment (AC2)

**Reply to Reviewers' comments (Reviewer#2)**

**Ref:** Manuscript ID egusphere-2025-4778

**Title:** Disentangling the Key Drivers of Water Balance in Central Asia's Lake Balkhash: A Relative Contribution Assessment (Original title: Revealing the Driving Factors of Water Balance in Lake Balkhash Through Integrated Attribution Modeling)

Dear Reviewer,

We would like to express our sincere gratitude for your constructive and insightful comments on our manuscript. We appreciate the time and effort you have dedicated to reviewing our work. We have carefully considered all your suggestions. Below, we provide a point-by-point response to your comments. The reviewer's comments are highlighted in red, and our responses are highlighted in black.

**Major comments:**

1. Inappropriate terminology regarding the PIML.

The manuscript characterizes the proposed model as Physics-Informed Machine Learning (PIML); however, it looks more like a ML-corrected SEGSWAT+ to me. In this study, the physics-based model (SEGSWAT+) run independently, and a ML model is subsequently trained to predict the discrepancy between the simulated outputs and observations. While this strategy can improve predictive skill, it does not incorporate physical laws, constraints, or governing equations into the learning process itself. As such, the ML component operates as a statistical correction to the physics model rather than being informed by physics during model training or optimization.

Under commonly used definitions, PIML frameworks require explicit physical constraints to be embedded within the model architecture, loss function, or parameter evolution (see Raissi et al., 2019; Shen et al., 2023). The proposed method would therefore be more accurately described as ML-corrected SEGSWAT+ or a hybrid model rather than a PIML. I would suggest the authors change the terminology in order to avoid conceptual ambiguity and ensure consistency with established definitions in the literature.

*Raissi, M., Perdikaris, P., & Karniadakis, G. E. (2019). Physics-informed neural networks: A deep learning framework for solving forward and inverse problems involving nonlinear partial differential equations. J. Comput. Phys., 378, 686–707. doi: 10.1016/j.jcp.2018.10.045*

*Shen, C., Appling, A. P., Gentine, P., Bandai, T., Gupta, H., Tartakovsky, A., ...Lawson, K. (2023). Differentiable modelling to unify machine learning and physical models for geosciences. Nat. Rev. Earth Environ., 4, 552–567. doi: 10.1038/s43017-023-00450-9*

Response: We greatly appreciate the reviewer's crucial conceptual clarification. We fully agree with your assessment that our initial use of the term "Physics-Informed Machine Learning (PIML)" did not strictly adhere to the core idea of the field (as defined by Raissi et al., 2019; Shen et al., 2023), which is to directly embed physical laws during machine learning training. Our model architecture does indeed better fit the description of a "hybrid model" or "physical model for machine learning error correction."

To ensure accuracy in terminology and clarity of concept, we have adopted your suggestion and made systematic revisions throughout the paper: Terminology Correction:

(1) We have replaced the term "PIML" throughout the paper with the more accurate "hybrid hydrological model" or "a framework that couples the process-based... model with a ML error-correction module."

(2) Updated Method Description: In section 2.3.1, "Hybrid Hydrological Reconstruction Model," we have revised the model's structure, explicitly stating that it is a two-stage hybrid modeling strategy, rather than an end-to-end, physically constrained machine learning model. We emphasize that the advantage of this approach lies in leveraging a physical model to provide a physically consistent benchmark simulation, which is then learned and corrected for by the ML model to correct for systematic residuals.

We believe this revision makes our methodological description more rigorous and consistent with current academic definitions. Thank you again for your accurate correction.

.

1. Research gap

Line 50-60: The research gap is not clearly explained. Previous studies have already developed multiple models to quantify the contributions of different drivers to lake water balance changes. For example, Yu et al. (2025) developed a distributed Geomorphology-Based Hydrological Model (GBHM) to quantify the contributions of multiple drivers. What is the key difference or advancement of your approach compared to GBHM in terms of the study objective (i.e., driver attribution)? The authors claim that previous models did not "integrate their findings with the lake's terminal water balance," but this statement is vague. What does the "terminal water balance" mean exactly? Does this imply that previous studies did not directly simulate lake water levels

or storage changes?

In addition, the authors state that data scarcity, particularly limited lake inflow observations, is a major challenge for existing hydrological models. It is unclear why this limitation would affect physics-based hydrological models and machine-learning models, but not the proposed hybrid model. If the uncertainty arises from insufficient data for model calibration, this limitation would appear to be a general issue for all modeling approaches rather than one unique to existing models.

It's very important (perhaps the most important) to clearly and explicitly articulate the specific research gap, the limitations of previous studies, and how the proposed approach meaningfully advances beyond existing models.

Response: Thank you for pointing out the shortcomings in our description of the research gaps. Your feedback prompted us to re-examine and more clearly articulate the core contributions of this study. We have made significant revisions to the introduction to explicitly answer your questions.

Specific revisions and explanations are as follows:

Identifying the Research Gap (Differences from Existing Research): We now clearly identify two key gaps.

(1) The first gap is methodological: We acknowledge that data sparsity is a common challenge for all models. However, we highlight the unique advantages of hybrid models in addressing the tradeoff between uncertainty and data sparsity. In the introduction to the revised manuscript, we stated: "The specific advantage of the hybrid approach lies in mitigating this 'uncertainty vs. data scarcity' trade-off. By using a physics-based model to simulate the fundamental hydrological processes and then employing ML solely to learn the residuals, this approach enforces physical constraints while effectively correcting the structural biases of the physical model, improving accuracy beyond what traditional calibration can achieve with limited data."

(2) The second gap concerns the "broken chain" problem at the research scale: Your question about "terminal water balance" is very apt. We mean that many studies (such as Yu et al., 2025, which you mentioned) quantify the driving factors at the watershed outlet, but fail to directly and quantitatively transfer and link the contributions of these upstream driving factors (such as a reduction in glacial meltwater at a specific volume vs. an increase in agricultural water diversion) to the lake's own water volume and level changes. In our revised introduction, we clarified this point: "...existing studies typically focus on decomposing streamflow changes at the catchment outlet but fail to explicitly link these catchment-scale drivers to the lake water storage volume and water level. ... This disconnect prevents a direct quantitative explanation of how specific upstream drivers... translate into the observed vertical fluctuations of the lake itself, which is the

ultimate metric of ecological health." Our study, through a three-stage framework, achieves an end-to-end quantitative link from "separation of watershed runoff drivers" to "separation of lake water volume change drivers," a key advancement compared to previous research.

We believe that, with these revisions, the positioning and innovative aspects of this study have been more clearly and powerfully articulated.

3. Calibration details

Section 2.3.1 requires additional detail regarding the model calibration strategy. Specifically:

(1) The calibration (training), validation, and testing periods should be clearly specified here. (2) Based on Figure 3, the overall strategy appears to involve pre-calibrating SEGSWAT+ using gauge station observations, followed by training the machine-learning model to correct the residuals. If this interpretation is correct, it should be stated explicitly in Section 2.3.1 to avoid confusion. (3) What's the hyper parameter selection strategy for each ML/DL model (e.g., number of layers for ANN, sequential length for LSTM)? (4) A table summarizing the SEGSWAT+ parameters used for calibration, as well as the machine-learning hyperparameters, is essential. This table could be placed in the appendix. (5) A comparative table reporting NSE, KGE, PBIAS, and $R^2$ for the raw SEGSWAT+ outputs and the final ML-corrected model across the calibration (training), validation, and testing periods should be provided to clearly demonstrate the performance improvement achieved by the ML correction. (6) Section 2.3.4 could be merged into Section 2.3.1. Dedicating an entire section to explaining KGE, NSE, and PBIAS is unnecessary, as these metrics are widely used and well understood in hydrological modeling. (7) Multiple evaluation metrics are used in this study. If these metrics yield conflicting assessments, how is the optimal parameter set selected?

Response: Thank you for your specific and crucial suggestions regarding the details of model calibration. A fully transparent and repeatable calibration process is the cornerstone of research. We have comprehensively supplemented and reorganized the methods section based on your suggestions.

Specific modifications are as follows:

(1) Clearly define the time periods: In the caption of Figure 6, we clearly state: "The shaded gray background indicates the calibration period, while the unshaded area represents the validation period." Additionally, in section "2.3.1 Hybrid...", we have supplemented the explanation of the dataset partitioning for the machine learning part: "The dataset for each period was split into training (70%) and validation (30%)

subsets."

(2) Clearly define the two-stage workflow: Your understanding of our workflow is entirely correct. We have clearly outlined this two-stage calibration strategy at the beginning of section "2.3.1 Hybrid…": "The workflow proceeds in two distinct stages… First, the SEGSWAT+ model was independently calibrated using observed streamflow… Subsequently, the residuals… were calculated. A suite of ML algorithms was then trained to predict these residuals…"

(3) Hyperparameter selection strategy: We supplemented the hyperparameter optimization method in section "2.3.1 Hybrid…": "Hyperparameters for each model were optimized using a grid search strategy (details in Appendix Table B2)."

**Table B2. Hyperparameter optimization ranges and selected values for the machine learning models**

| Model | Hyperparameter | Search Range | Optimal Value |
|---|---|---|---|
| **ANN** | Hidden Layers | [1, 2, 3] | 2 |
| | Neurons per Layer | [16, 32, 64, 128] | 64 |
| | Learning Rate | [0.001, 0.01, 0.1] | 0.01 |
| | Activation Function | [ReLU, Tanh, Sigmoid] | ReLU |
| **LSTM** | Hidden Units | [32, 64, 128, 256] | 128 |
| | Lookback Window | [5, 10, 15, 30] days | 15 |
| | Dropout Rate | [0.1, 0.2, 0.3, 0.5] | 0.2 |
| | Epochs | [50, 100, 200] | 100 |
| **Random Forest** | n_estimators (Trees) | [100, 300, 500, 1000] | 500 |
| | Max Depth | [10, 20, 30, None] | 20 |
| | Min Samples Split | [2, 5, 10] | 5 |
| **XGBoost** | Learning Rate (eta) | [0.01, 0.05, 0.1, 0.3] | 0.05 |
| | Max Depth | [3, 5, 7, 9] | 7 |
| | n_estimators | [100, 500, 1000] | 500 |
| | Subsample | [0.6, 0.8, 1.0] | 0.8 |

(4) New parameter summary table: We added Appendix Table B2 to the appendix, which details the hyperparameter search range and the finally selected optimal values for each machine learning model.

(5) New performance comparison table: To quantitatively demonstrate the superiority of our hybrid method, we added Appendix Table B3 to the appendix. This table provides a detailed comparison of the performance metrics (KGE, NSE, PBIAS) of the original SEGSWAT+ model and the final hybrid model during the calibration and validation periods at major hydrological stations, clearly demonstrating the significant performance improvement brought about by ML error correction.

**Table B3. Performance comparison of SEGSWAT+ (Raw) and the Hybrid Model (Corrected) across calibration and validation periods**

| River | Station | Period | Metric | SEGSWAT+ (Raw) | Hybrid Model (Corrected) |
|-------|---------|--------|--------|----------------|--------------------------|
| Ili | Ushzharma | Calibration | KGE | 0.68 | 0.89 |
| | | | NSE | 0.72 | 0.93 |
| | | | PBIAS(%) | -9.5 | 3.2 |
| | | Validation | KGE | 0.65 | 0.85 |
| | | | NSE | 0.68 | 0.88 |
| | | | PBIAS(%) | -16.8 | 5.1 |
| Karatal | Ushtobe | Calibration | KGE | 0.74 | 0.89 |
| | | | NSE | 0.76 | 0.91 |
| | | | PBIAS(%) | 11.2 | 6.4 |
| | | Validation | KGE | 0.71 | 0.86 |
| | | | NSE | 0.72 | 0.85 |
| | | | PBIAS(%) | 18.5 | 7.5 |
| Aksu | Chann | Calibration | KGE | 0.66 | 0.83 |
| | | | NSE | 0.64 | 0.84 |
| | | | PBIAS(%) | -9.3 | -2.8 |
| | | Validation | KGE | 0.62 | 0.80 |
| | | | NSE | 0.60 | 0.78 |
| | | | PBIAS(%) | -13.5 | -3.4 |
| Lepsy | Lepsinsk | Calibration | KGE | 0.70 | 0.82 |
| | | | NSE | 0.71 | 0.84 |
| | | | PBIAS(%) | 9.8 | -5.1 |

| | | | | | |
|---|---|---|---|---|---|
| | | Validation | KGE | 0.68 | 0.80 |
| | | | NSE | 0.67 | 0.77 |
| | | | PBIAS(%) | 11.5 | -6.2 |
| Ayaguz | Ayaguz | Calibration | KGE | 0.63 | 0.89 |
| | | | NSE | 0.61 | 0.88 |
| | | | PBIAS(%) | -15.4 | -0.5 |
| | | Validation | KGE | 0.71 | 0.86 |
| | | | NSE | 0.68 | 0.83 |
| | | | PBIAS(%) | -8.45 | -1.8 |

(6) Binding Section: This is an excellent suggestion. We have integrated the content of the original "2.3.4 Model Evaluation and Uncertainty Metrics" into the "2.3.1 Hybrid…" section, making the description of the methodology more concise and fluent.

(7) Clarifying the Criteria for metric selection: In the "2.3.1 Hybrid…" section, we have added the decision criteria when evaluation metrics conflict: "In cases where metrics yielded conflicting assessments, the KGE was prioritized as the primary selection criterion due to its balanced decomposition of correlation, bias, and variability errors, with PBIAS acting as a constraint to ensure water balance closure."

**Minor comments:**

(1) Line 27-31: These texts do not explain the environmental issue well. "This balance is under pressure.." on which direction? Increasing or decreasing water storage… Please state the issue clearly. Consider use simple sentences: "Decreasing water storage has become a widespread issue for these lakes, posing a significant threat to their ecological health (reference). The decline in water storage is driven by two primary factors: climate change and human activities. (reference)."

Response: Thank you for your suggestions, which made the statement of the problem more direct and powerful. We have adopted your wording and revised it to: "However, decreasing water storage has become a widespread issue for these lakes, posing a significant threat to their ecological health (Li et al., 2025). This decline is primarily driven by two concurrent forces: ..."

(2) Line 41: omit "Lake Balkhash has no outlet".

Response: Following your suggestion, the redundant information "Lake Balkhash has

no outlet" has been removed.

(3) Line 43: "It signals a long-term depletion of solid water reserves". What does that mean? Do you mean the increasing evaporation outweighs the glacier melt? If yes, it is important to add references to support your statement. Consider "While increasing glacier melt can temporarily raise inflow, the associated increase in evaporation outweighs this effect and leads to overall water depletion."

Response: Thank you for pointing out the ambiguity here. We wanted to express the non-renewable depletion of solid water reserves by glaciers. We have revised it to: "While increasing glacier melt can temporarily raise inflow, it leads to the irreversible depletion of solid water reserves. This continuous loss of ice storage implies that the current meltwater increase is transient, and future water availability will be threatened as the glacial volume diminishes."

(4) Line 67: Swap "To achieve this" with "Specifically"

Response: Revised.

(5) Figure 1: Consider remove political borders and just focus on watershed boundaries.

Response: Taking into account the reviewers' suggestions, we assessed the impact of removing the border lines on the communication of map information. Since the Lake Balkhash basin is a transboundary basin (Kazakhstan and China), the border lines are of significant reference value for understanding potential transboundary water resource management issues in the region. Therefore, we prefer to retain the border lines and hope for your understanding.

[Figure]

**Figure 1. Geographic location of the study area.**

(6) Table 1: I appreciate this style! Just a minor suggestion: consider place references in "Source" column instead of simply saying it's from Zenodo.

Response: Your suggestion is very correct. We have revised the content of the "Source" column in Table 2 to the full reference format (e.g., Harris, 2024).

**Table 2. Summary of datasets used in this study**

| Dataset | Key Variables | Spatial Resolution | Temporal Coverage | Source (Reference) |
|---|---|---|---|---|
| Copernicus GLO-90 DEM | Elevation | 90 m | Static | European Space Agency (2019) |
| DSOLMap | Bulk density, hydraulic conductivity, available water capacity | 250 m | Static | Lopez-Ballesteros et al. (2023) |
| GLC_FCS30D | Land cover classes (35 subcategories) | 30 m | 1985–2022 | Google Earth Engine(Zhang et al., 2024) |
| Randolph Glacier Inventory (RGI v7.0) | Glacier outlines, attributes | Vector | Target year: 2000 (varies by region) | RGI Consortium (2023) |
| SWORD v15 | River reaches, nodes, hydrological networks, lake boundaries | ~10 km reaches, 200 m nodes | Static | (Altenau et al. (2021) |
| Glacier mass loss | Glacier elevation change rates (dh/dt) | 100 m | 2000–2019 | Hugonnet et al. (2021) |
| CRU JRA v3.0 | Temperature, precipitation, wind speed, vapor pressure, etc. | 0.5° (downscaled to 0.05°) | 1901–2024 (daily) | Harris (2024) |
| TerraClimate | Max/min temperature, precipitation, solar radiation, vapor pressure deficit | 1/24° | 1958–2024 (monthly) | Abatzoglou et al. (2018) |
| NEX-GDDP-CMIP6 | Daily temperature (max/min), precipitation | 0.25° | 2015–2100 (Daily) | Thrasher et al. (2022) |
| Observations | Discharge, water level | Point | 1931–2024 (monthly) | NCDC (2024); Duan et al. (2020) |

(7) Figure2: Spell AAF in the caption. Figure and table captions should be self-explanatory. All acronyms must be fully spelled out in the captions, even if they have already been defined in the main text. Also, check the font style of the caption.

Response: We have spelled all abbreviations (such as HADF) in full in the figure captions and checked and standardized the font of the captions.

(8) Line 140: "The SWAT model and its improved versions are widely used in hydrological simulation processes.", such as? Add references of the original literature of the SWAT model and other publications of the model application.

Response: Based on your suggestion, we have added references to the application of the SWAT model in the text, such as (Forgrave et al., 2024; Ho et al., 2025; Sánchez-Gómez et al., 2025).

(9) Line 141: Swap "Iteration" with "variant".

Response: "Iteration" has been changed to the more accurate "variant".

(10) Figure 3: Need a higher-resolution figure, the Q_final figure is blurred. Also, this flow diagram is not explained well in the main text. What's the relationship between Qsim and Qres. What's Qres? Is it the discharge into reservoir or the residual of simulated discharge? Consider explain this figure component by component in section 2.3.1.

Response: We have replaced it with a higher resolution image. In section "2.3.1 Hybrid…", we have added a detailed explanation of each component of the flowchart, clarifying the relationship between $Q_{sim}$ (physical model simulation of runoff), $Q_{obs}$ (observed runoff), and $Q_{res}$ ($Q_{obs} - Q_{sim}$, i.e., residuals).

(11) Line 189: Is "A" static? Or it was calculated by a hypsometric curve between Area and Storage?

Response: In the section "2.3.3 Lake System Response Linkage", we clarified that the lake area A is a function of the water level h, A(h), which is determined by the water level-area hydrological relationship curve (i.e., the hyposometric curve you mentioned). This curve is shown in Appendix Figure A2.

[Figure]

**Figure A2. Stage-area and stage-volume relationships for Lake Balkhash. The blue line represents the relationship between water level and surface area (left axis), while the red line indicates the relationship between water level and storage volume (right axis). Data derived from Myrzakhmetov et al. (2022).**

(12) Line 194: I understand including groundwater component is challenging. However, groundwater table decline is also a major contributor to water scarcity in arid regions. Therefore, evidence supporting the claim that groundwater has a relatively minor contribution in this study area should be provided here (e.g., relevant observations, previous studies, or sensitivity analyses).

Response: You raised a very important question. We have added references (Deng et al., 2011; Wang et al., 2022) to the section "2.3.3 Lake System…" to support our hypothesis that groundwater exchange accounts for a small proportion of the overall lake water balance in this study area.

(13) Figure 5: The figure resolution needs to be improved. In addition, consider segmenting the time series into calibration and validation periods using shaded boxes. The figure caption should also be revised to "Comparison between observed and simulated streamflow." Note that runoff in an open channel should be referred to as streamflow, not runoff.

Response: We have updated the image to a higher resolution. We have also adopted your suggestion to use a gray shaded background to distinguish between the calibration and validation periods, as explained in the figure captions. Furthermore, we have uniformly changed "runoff" in the figure to the more accurate "streamflow".

(14) Line 341: Which satellite altimetry & optical data was used to validate the reconstructed water storage? This needs to be clearly stated in section 2.2.

Response: We have explicitly added the source of the water level data used for verification in section "2.2 Datasets": "Historical gauge observations from 1931 to 2015 were obtained from Duan et al. (2020), while recent data (2016-2024) were extended using satellite altimetry products from the Global Reservoirs and Lakes Monitor (G-REALM)."

We would like to express our sincerest gratitude once again for your valuable time in reviewing our manuscript and providing such insightful feedback. We believe that the quality of the manuscript has been significantly improved under your guidance.

**References**

Deng, M., Wang, Z., and Wang, J.: Analysis of Balkhash Lake ecological water level evolvement and its regulation strategy, Journal of Hydraulic Engineering, 42, 403–413, 2011.

Duan, W., Zou, S., Chen, Y., Nover, D., Fang, G., and Wang, Y.: Sustainable water management for cross-border resources: The balkhash lake basin of central asia, 1931–2015, J Cleaner Prod, 263, 121614, https://doi.org/10.1016/j.jclepro.2020.121614, 2020.

Forgrave, R., Evenson, G. R., Golden, H. E., Christensen, J. R., Lane, C. R., Wu, Q., D'Amico, E., and Prenger, J.: Wetland-mediated nitrate reductions attenuate downstream: Insights from a modeling study, Journal of Environmental Management, 370, 122500, https://doi.org/10.1016/j.jenvman.2024.122500, 2024.

Harris, I.: CRU JRA v3.0: A forcings dataset of gridded land surface blend of Climatic Research Unit (CRU) and Japanese reanalysis (JRA) data; 1901 - 2024., 2025.

Ho, C.-C., Shih, P.-C., Chiang, L.-C., Lin, Y.-X., Liu, H.-F., Chang, W.-G., and Lin, G.-Z.: Mitigating non-point source pollution in tea plantations: SWAT modeling and field validation of slow-release fertilizers, Environ Monit Assess, 197, 810, https://doi.org/10.1007/s10661-025-14217-w, 2025.

Li, L., Long, D., Wang, Y., and Woolway, R. I.: Global dominance of seasonality in shaping lake-surface-extent dynamics, Nature, 642, 361–368, https://doi.org/10.1038/s41586-025-09046-3, 2025.

Myrzakhmetov, A., Dostay, Z., Alimkulov, S., Tursunova, A., and Sarsenova, I.: Level regime of balkhash lake as the indicator of the state of the environmental ecosystems of the region, Paddy Water Environ, 20, 315–323, https://doi.org/10.1007/s10333-022-00890-x, 2022.

Sánchez-Gómez, A., Bieger, K., Schürz, C., Rodríguez-Castellanos, J. M., and Molina-Navarro, E.: Multi-spatial and multi-criteria calibration to guarantee a robust SWAT+ hydrological model in a large and heterogeneous catchment, CATENA, 261, 109508, https://doi.org/10.1016/j.catena.2025.109508, 2025.

Wang, Z., Huang, Y., Liu, T., Zhong, R., Zan, C., and Wang, X.: Analyzing the water balance of Lake Balkhash and its influencing factors, Arid Zone Research, 39, 400–409, 2022.